# Structural basis for calcium-stimulating pore formation of *Vibrio* α-hemolysin

Yu-Chuan Chiu[1,7], Min-Chi Yeh[2,7], Chun-Hsiung Wang[2], Yu-An Chen[1], Hsiang Chang [1], Han-You Lin[3], Meng-Chiao Ho [2,4,5] & Shih-Ming Lin [1,6] ✉

*Vibrio* α-hemolysins (αHLs) are β-pore-forming toxins secreted by *Vibrio* pathogens, crucial for the facilitation of bacterial infections through host cell lysis. These toxins are produced as inactive precursors, requiring proteolytic maturation and membrane association for activation within host tissues. Here, we investigate *Vibrio campbellii* αHL (VcαHL), and establish that its hemolytic activity is significantly stimulated by calcium ions, with an $EC_{50}$ that aligns with physiological calcium concentrations. Furthermore, we illustrate the vital contribution of calcium ions to the oligomerization of VcαHL on membranes. Using X-ray crystallography and cryo-electron microscopy, we decipher both the immature and assembled structures of VcαHL and elucidate the conformational changes corresponding to toxin assembly. We also identify a calcium-binding module that is integral for VcαHL's calcium-dependent activation. These findings provide insights into the regulatory mechanisms of VcαHL and have the potential to inform the development of targeted therapeutic strategies against *Vibrio* infections.

Several pathogenic *Vibrio* species, inclusive of *Vibrio cholerae*, *Vibrio parahaemolyticus*, and *Vibrio vulnificus*, secrete α-hemolysins (αHLs) as pivotal virulence factors to invade into host tissues[1–4]. These αHLs, classified under the β-pore-forming toxins (β-PFTs) family, are capable of oligomerizing into heptameric ring-shaped assemblies and forming transmembrane pores on host cell membranes[3–6]. Typically, *Vibrio* αHLs are secreted as water-soluble monomers, which are activated via proteolytic cleavage of a specific protective domain. Upon activation, these αHLs oligomerize to create pre-pore complexes on target cell membranes, consequently forming β-barrel transmembrane pores that penetrate the membrane lipid bilayer. These resultant pores can catalyze the destruction of erythrocytes, compromise host tissues, and facilitate the evasion of host immune responses[5,7]. The oligomerization process is tightly regulated to prevent self-inflicted damage prior to secretion and to enable a more effective and targeted attack on the host cells. A comprehensive understanding of the regulatory mechanisms of *Vibrio*

αHLs is essential to the development of novel therapeutic strategies and to enhance our knowledge of complex host-pathogen interactions.

Several *Vibrio* αHLs have been identified and intensely investigated to comprehend their mechanism of virulence[4]. Notably, *V. cholerae* αHL, also termed as *V. cholerae* cytolysin (VCC), is one of the most thoroughly characterized toxins in this group[8–10]. The structures of VCC, in both its monomeric and oligomeric conformations, have been determined[11,12]. VCC consists of multiple distinct regions, including a pro domain, a cytolysin domain, a pre-stem loop, and uniquely, a β-trefoil lectin domain and a β-prism lectin domain. These lectin domains, which exhibit structural resemblances to carbohydrate-binding domains of plant lectins like ricin and jacalin, are a rarity among bacterial β-PFTs[13,14]. The pro-domain functions as a protective measure, inhibiting αHL activation and necessitating proteolytic cleavage before launching an attack on host membranes. The cytolysin domain facilitates oligomerization into heptameric structures and binds

[1]Department of Biotechnology and Bioindustry Sciences, National Cheng Kung University, Tainan, Taiwan. [2]Institute of Biological Chemistry, Academia Sinica, Taipei, Taiwan. [3]Department of Veterinary Medicine, School of Veterinary Medicine, National Taiwan University, Taipei, Taiwan. [4]Institute of Biochemical Sciences, National Taiwan University, Taipei, Taiwan. [5]Graduate Institute of Biochemistry and Molecular Biology, National Taiwan University, Taipei, Taiwan. [6]Institute of Tropical Plant Sciences and Microbiology, National Cheng Kung University, Tainan, Taiwan. [7]These authors contributed equally: Yu-Chuan Chiu, Min-Chi Yeh. ✉e-mail: smlin@mail.ncku.edu.tw

the pre-stem loop until the formation of the transmembrane domain. The β-trefoil lectin domain is fundamental for maintaining protein folding and overall toxin functionality[15], while the β-prism lectin domain assists in binding polysaccharides or glycan structures on host cell membranes[16–18]. Notably, the β-prism lectin domain exhibits a predilection for specific types of carbohydrates, which could potentially be associated with the selective host cell targeting strategy of the toxin[16].

Besides binding to membrane-associated glycans, the presence of cholesterol and sphingolipids within the lipid bilayer also play a critical role in enhancing toxin assembly and pore formation by *Vibrio* αHL[9,19,20]. Cholesterol has been found to modulate *Vibrio* αHL activity through a direct interaction with the toxin molecule rather than simply modifying the physicochemical properties of the target membrane[19,21]. Despite these findings, the precise cholesterol binding site of *Vibrio* αHL remains unidentified. Interestingly, previous research demonstrated that deletion of the transmembrane stem loops dose not impede pre-pore formation in VCC, implying that membrane integration is not a prerequisite for toxin oligomerization[22]. Therefore, the mechanism regulating pre-pore assembly of *Vibrio* αHL prior to membrane integration still demands further investigation.

In this work, we characterize an αHL from a marine pathogen, *Vibrio campbellii*, herein referred to as VcαHL, to better understand the regulatory mechanism of *Vibrio* αHL pore formation processes.

Although VcαHL shares sequence conservation with well-characterized *Vibrio* αHLs, such as VCC and *V. vulnificus* hemolysin (VVH) (Supplementary Fig. 1, Supplementary Table 1), we report a unique attribute of VcαHL–its calcium-dependent oligomerization activity. The presence of calcium ions significantly stimulated VcαHL's hemolytic activity and membrane permeability. Through determination of the protein structures using X-ray crystallography and cryo-electron microscopy (cryo-EM), we identified the calcium binding site in assembled VcαHL. Remarkably, mutations at the calcium binding residues nullified the calcium dependency of VcαHL, emphasizing the critical role of calcium binding in regulating the activation of toxin assembly. These findings elucidate the regulatory mechanism of the toxin assembly in *Vibrio* αHLs.

## Results

### Role of calcium ions in stimulating the toxin assembly and hemolytic activity of VcαHL

To elucidate the regulatory mechanisms of *Vibrio* αHL activity, we utilized an *Escherichia coli* system to heterologously express and purify the precursor form of *V. campbellii* αHL (pro-VcαHL), containing its N-terminal pro domain. This purified pro-VcαHL exhibited a molecular weight of 75 kDa in SDS-PAGE and could be transformed into its mature 65 kDa form via limited trypsinolysis treatment (Fig. 1a). Similar

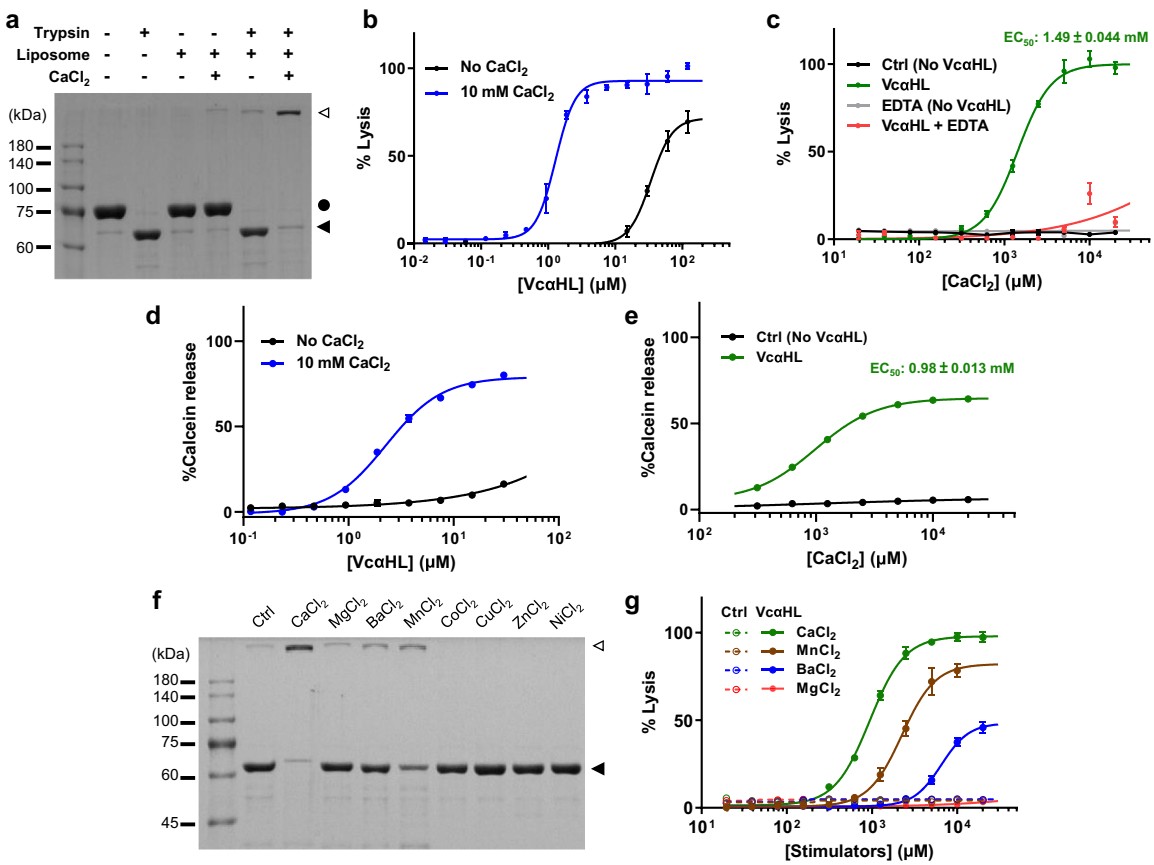

**Fig. 1 | Calcium ions stimulate the pore formation and hemolytic activity of VcαHL. a** SDS-PAGE analysis of pro-VcαHL (filled circle, 81 kDa) after it was treated with trypsin, liposomal membranes, and CaCl₂, resulting in mature VcαHL (filled triangle, 65 kDa) and SDS-resistant oligomers (open triangle). This experiment was conducted three times independently, yielding consistent results. **b** The hemolytic activity of VcαHL against erythrocytes was measured at various concentrations of VcαHL with or without CaCl₂. **c** The hemolytic activity of 7.5 μM VcαHL was measured at different CaCl₂ concentrations, both in the presence and absence of 10 mM EDTA. The 'Ctrl' group, with no added VcαHL, served as the control. **d** Calcein release from liposomes was evaluated at varying VcαHL concentrations with or

without 10 mM CaCl₂. **e** Calcein release assay was conducted at different CaCl₂ concentrations using 7.5 μM VcαHL. **f** SDS-PAGE analysis of VcαHL oligomerization in the presence of various divalent cations was performed. Three independent experiments showed similar results. Filled and open triangles indicate monomeric and oligomeric VcαHL, respectively. **g** Hemolytic activity of VcαHL in the presence of different divalent cations was measured. Ctrl represents the control group without VcαHL addition. All data points represent the means of three replicates, with error bars indicating standard deviation (S.D.). Curves depict dose-response fitting results using non-linear regression.

to VCC, the mature VcαHL has the capacity to self-assemble into transmembrane pore complexes in the presence of liposomal membranes[11]. These complexes are stable β-barrel transmembrane pores that could form high-molecular weight products resistant to denaturation by SDS (Fig. 1a). Remarkably, the addition of calcium ions significantly enhanced the toxin assembly and pore formation of mature VcαHL. The ratio of the assembled to monomeric VcαHL was notably higher in the calcium-ion treated group compared to the untreated group (Fig. 1a).

The role of calcium ions in VcαHL pore formation was further assessed by measuring its hemolytic activity against erythrocytes in the presence of calcium ions. Remarkably, the $EC_{50}$ value of VcαHL decreases to 1.29 μM compared to 33.84 μM (a 26-fold decrease) upon the addition of 10 mM $CaCl_2$ (Fig. 1b). This suggests that calcium ions indeed stimulate VcαHL's hemolytic activity. To obtain a more accurate understanding of the $Ca^{2+}$ efficacy in the VcαHL stimulation, we measured the hemolytic activity of VcαHL at a protein concentration of 7.5 μM that was significantly stimulated by calcium ions. This evaluation was conducted across a calcium concentration ranging from 0.02 mM to 20 mM. Our results further confirmed that $Ca^{2+}$ stimulated VcαHL's toxin function in a dose-dependent manner with an $EC_{50}$ value of 1.49 mM (Fig. 1c). In addition, presence of 10 mM EDTA would dramatically eliminate the hemolysis activity of VcαHL stimulated by calcium ions (Fig. 1c). The $EC_{50}$ levels corresponding to serum calcium ion concentrations indicate the physiological relevance of VcαHL's stimulation in the host environment[23]. Additionally, VcαHL demonstrated efficient calcein release from liposomes in the presence of calcium ions, further accentuating the calcium stimulatory effect (Fig. 1d, e).

The ion specificity for VcαHL assembly stimulation was explored by assessing the effects of various divalent cations. Both $Ba^{2+}$ and $Mn^{2+}$ were found to slightly enhance the pore formation of VcαHL, unlike $Mg^{2+}$, $Zn^{2+}$, $Co^{2+}$, and $Cu^{2+}$ which did not exhibit a similar effect (Fig. 1f). In hemolytic assays, $Ba^{2+}$ and $Mn^{2+}$ displayed stimulatory effects on VcαHL, albeit with higher $EC_{50}$ values compared to $Ca^{2+}$. In contrast, $Mg^{2+}$ did not enhance VcαHL's hemolytic activity (Fig. 1g). These findings strongly suggest that VcαHL functions as a $Ca^{2+}$-dependent hemolysin, with ion stimulation specificity limited to certain cations.

## The interplay of calcium and cholesterol in enhancing VcαHL assembly occurs through distinctive mechanisms

Previous research has highlighted the influential role of cholesterol content in target membranes on the hemolytic activity of several *Vibrio* αHLs[9,19,20]. Given these findings, we explored the possibility that the calcium-stimulated activity of VcαHL might intersect with this cholesterol-dependent regulation. To investigate this, we designed experiments to assess VcαHL toxin assembly in liposomal membranes, altering the cholesterol content (0% and 10%) and the presence of $Ca^{2+}$. Our observations revealed a higher assembly efficiency of VcαHL in cholesterol-rich membranes relative to their cholesterol-free counterparts (Fig. 2a). Interestingly, VcαHL exhibited a modest capacity to oligomerize into complexes even in cholesterol-free membranes when $Ca^{2+}$ was present.

To gain a more comprehensive understanding of the roles of these stimulating agents in VcαHL assembly, we employed negative stain transmission electron microscope (TEM) to visualize VcαHL oligomerization on liposomal membranes. We observed VcαHL forming numerous ring-shaped oligomers on membranes containing 10% cholesterol when $Ca^{2+}$ was present (Fig. 2b). Even more intriguing was that in cholesterol-free conditions, VcαHL managed to form a comparable number of oligomers, provided that calcium ions were presence. This suggests that $Ca^{2+}$ independently promotes VcαHL oligomerization, irrespective of the presence of cholesterol. Given that the ring-shaped oligomers formed on cholesterol-free membranes did not resist SDS denaturation (Fig. 2a), we hypothesize that these SDS-

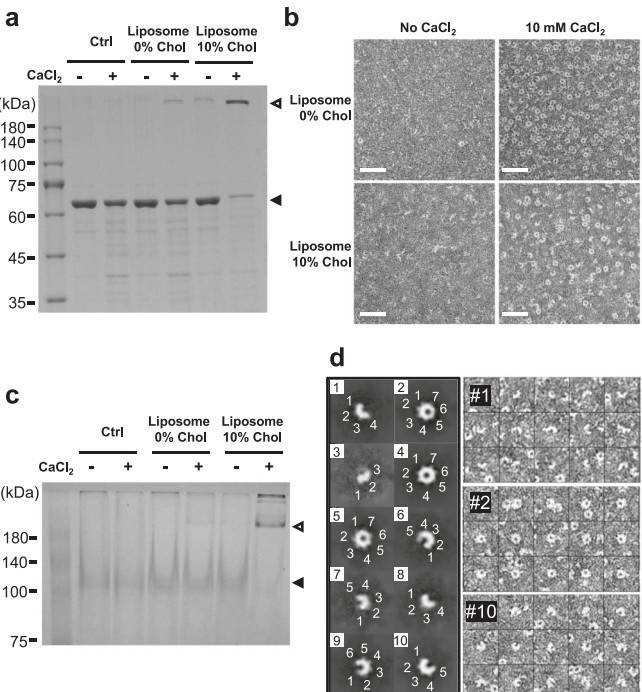

**Fig. 2 | $Ca^{2+}$ enhances the oligomerization of heptameric VcαHL on liposomal membranes. a** SDS-PAGE analysis of mature VcαHL assembled on liposomal membranes, with or without the inclusion of 10% cholesterol, and in the presence or absence of $Ca^{2+}$. Filled triangles denote unassembled VcαHL, whereas open triangles indicate assembled VcαHL. Control groups encompass VcαHL proteins that were not subjected to liposomal membrane treatment. This experiment was repeated three times independently with similar results. **b** Negative-stain TEM provides insights into the oligomerization of VcαHL on liposomal membranes, which were presented with or without 10% cholesterol. At least 10 independent micrographs for each condition gave similar results. A scale bar of 50 nm is used for reference. **c** Blue native PAGE (BN-PAGE) analysis was performed to study the assembly of mature VcαHL on liposomal membranes under different conditions. Before the BN-PAGE analysis, the liposomal membrane was solubilized using 1% β-OG. Filled triangles represent unassembled VcαHL, while open triangles signify assembled VcαHL. The control groups consist of VcαHL proteins that were not treated with liposomal membrane. Four independent experiments yielded similar findings. **d** The left panel displays 2D classification clusters of VcαHL assembled on cholesterol-containing membranes with calcium stimulation. The white numerals within each class image designate the protomers in each respective oligomer. The right panels present original particle images for classes #1, #2, and #10.

sensitive ring-shaped oligomers have not yet fully assembled into transmembrane pores. Our interpretation speculates that calcium ions facilitate the formation of these VcαHL ring-shaped complexes on the membrane surface.

To further investigate these $Ca^{2+}$-stimulated VcαHL oligomers, we solubilize these oligomerized VcαHL from liposomal membranes by using 1% n-octyl-β-D-glucoside (β-OG) for blue native PAGE analysis. The β-OG is a mild detergent which is previously used for structural determination of VCC pores[12]. The results showed that calcium ions stimulate the oligomerization of VcαHL on both cholesterol-free and cholesterol-containing liposomal membranes (Fig. 2c). However, the oligomers assembled in cholesterol-free liposomes were found to be less abundant compared to those in cholesterol-containing liposomes. This suggests that the absence of membrane support could impact the subunit association of VcαHL complexes.

In addition, we noticed several partial assembled structures of VcαHL in the negative stain micrographs (Fig. 2b). Therefore, we analyzed single-particle images of VcαHL assembled on cholesterol-containing membranes with calcium stimulation. The 2D classification

results showed that the ring-shaped VcαHL composed of seven subunits, which is a conserved feature across *Vibrio* αHLs. Interestingly, we also identified oligomeric states ranging from trimers (3-mers) to hexamers (6-mers) as 'arc' structures (Fig. 2d), which indicate intermediate stages in the assembly process. It is important to note, however, that these intermediate oligomers were not detected in either SDS-PAGE or blue native PAGE analyses, suggesting that these intermediates were not yet integrated into membranes for constructing stable β-barrel structures. Thus, the association with the membrane might be a critical factor in maintaining the oligomeric structure of these intermediates. These findings suggest a crucial role of cholesterol content in the membrane for the formation of transmembrane pores by VcαHL. Simultaneously, they highlight the pivotal role of calcium ions in enhancing the oligomerization of VcαHL on the membrane surface.

### Crystal structure of pro-VcαHL revealed the unique conformation at β-lectin domains

To further explore the mechanism of calcium-mediated stimulation of VcαHL, we determined the crystal structure of pro-VcαHL (Supplementary Table 2). Paralleling the structure of pro-VCC, this structure also shows four distinct domains: a pro domain, a cytolysin domain, a β-trefoil lectin domain, and a β-prism lectin domain (Fig. 3a)[11]. However, the β-prism lectin domain of pro-VcαHL is situated at a distinct position, forming a unique conformation. In pro-VcαHL, the β-prism lectin domain primarily interacts with the β-trefoil lectin domain, contrasting with its binding to the pre-stem loop in the pro-VCC structure (Fig. 3b, c)[11]. Despite these differences, the distinct β-lectin domains in pro-VcαHL and pro-VCC maintain highly conserved

molecular structures (Supplementary Fig. 2a, b), suggesting that conformational differences arise due to individual domain movements. Notably, the N-terminal region of the pro domain in pro-VcαHL exhibits an alpha helix absent in the pro-VCC structure (Supplementary Fig. 2c), the functional implications of which remain to be investigated.

Several hydrogen bonds and salt bridges were formed between the β-prism and β-trefoil lectin domains, stabilizing the domain interactions in pro-VcαHL (Fig. 3d). In addition, the pre-stem loop in pro-VcαHL interacts with the β-trefoil lectin and cytolysin domains through several polar interactions (Fig. 3e). Two lysine residues, K300 and K302, interact with the β-trefoil lectin domain through side-chain associations with T534 and E529, respectively. In addition, a loop (L188-N199), also known as the cradle loop due to its role in cradling the pre-stem loop in pro-VCC[12,24], is also observed to interact with the pre-stem loop in VcαHL (Fig. 3e). Multiple residues at the interface between the pre-stem loop and the cytolysin domain engage in the formation of hydrogen bonds and salt bridges (Fig. 3f). These interactions are critical for anchoring the pre-stem loop to the cytolysin domain prior to transmembrane pore formation.

### Cryo-EM structure of assembled VcαHL reveals its conformational changes after oligomerization

To understand the structural transformations post-oligomerization, we subjected the assembled VcαHL to cryo-EM structural analysis. The VcαHL was assembled on liposomal membranes and solubilized using styrene-maleic acid (SMA), yielding SMA lipid particles (SMALPs) containing assembled VcαHL. Cryo-micrographs of VcαHL in SMALPs were acquired and analyzed for single-particle 3D reconstruction (Supplementary Fig. 3), resulting in a density map of 2.06 Å resolution

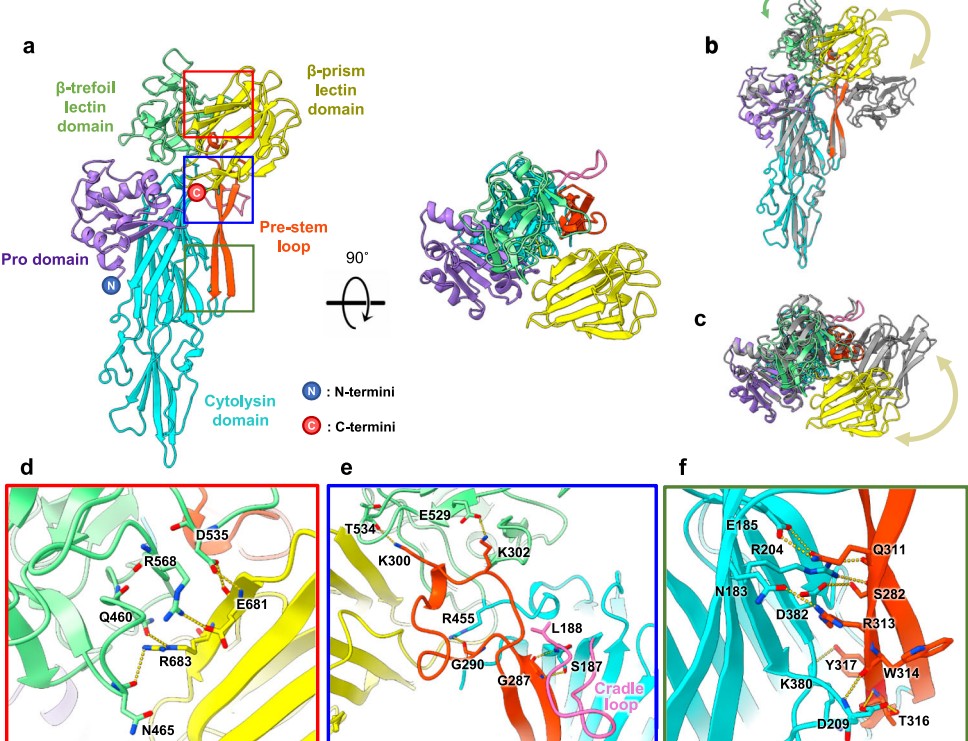

**Fig. 3 | Crystal structure of pro-VcαHL reveals a distinct binding site for the β-prism lectin domain. a** The overall structure of pro-VcαHL is displayed with each domain labeled and colored. The N and C indicate the N- and C-terminus, respectively. The colored boxes highlight the zoomed regions shown in panels **d**–**f**. The side view (**b**) and top view (**c**) of pro-VcαHL superimposed with pro-VCC (gray, PDB 1XEZ). Their structural differences in the β-trefoil and β-prism lectin domains are indicated by green and yellow curves, respectively. **d** The interactions between the β-trefoil and β-prism lectin domains are displayed, with interacting residues shown as sticks and labeled. Yellow dashed lines represent the hydrogen bonds connecting the two domains. The pre-stem loop is shown binding to (**e**) the β-trefoil lectin domain, the cradle loop (pink), and (**f**) the cytolysin domain through multiple hydrogen bonds (yellow dashed lines). The interacting residues are depicted as sticks and labeled.

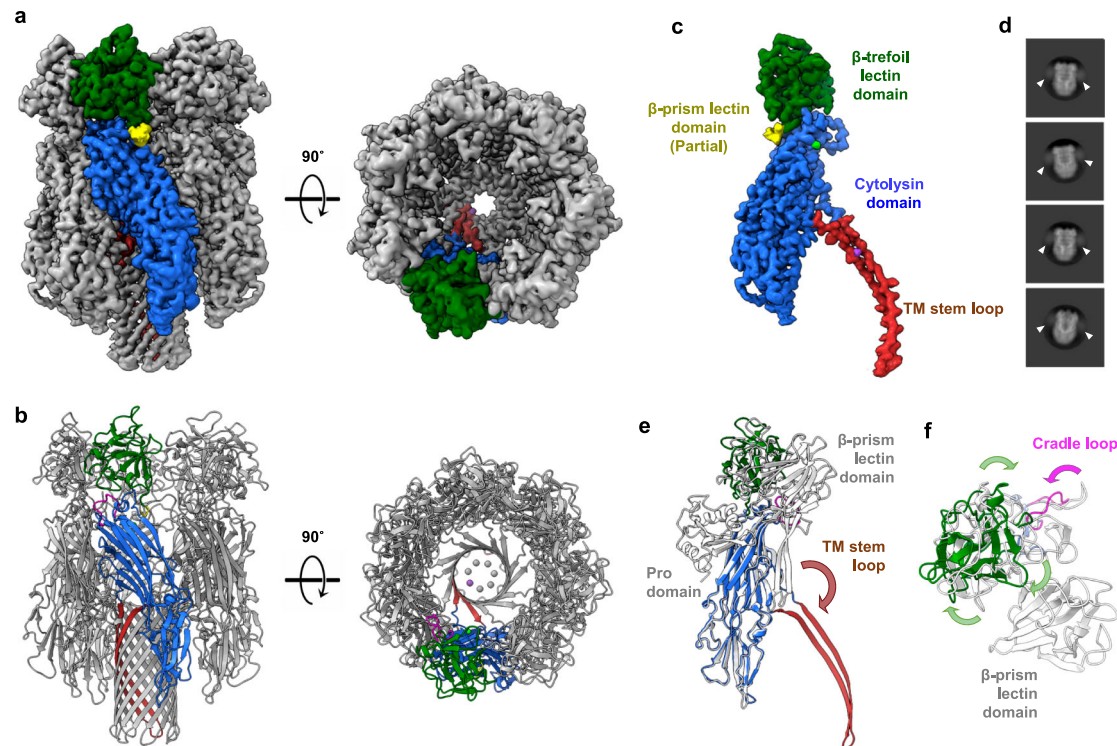

**Fig. 4 | Cryo-EM structure of the assembled VcαHL complex reveals conformational changes associated with transmembrane pore formation. a** The cryo-EM density map and **b** the coordinate model of the assembled VcαHL in SMALP are depicted in two orthogonal views, with one of the seven protomers colored by domain. **c** The protomer of assembled VcαHL is shown with each domain labeled and marked in different colors. The light green and purple densities indicate calcium and potassium ions, respectively. **d** Representative 2D classification images of the assembled VcαHL single particle images. The fuzzy densities of β-prism lectin domains were indicated by filled triangles. **e** The superimposed structures of the pro-VcαHL protomer (shown uncolored) and the assembled VcαHL (color-coded by domain). The brown arrow indicates movement of transmembrane (TM) stem loops after assembly. **f** A zoomed extracellular view of figure e illustrates the conformational changes at the β-trefoil lectin domain and the cradle loop, with arrows indicating the rotation direction of each region from pro-VcαHL to assembled VcαHL.

(Supplementary Table 3, Supplementary Fig. 4). The map revealed a heptameric complex exhibiting C7 symmetry, featuring a β-barrel transmembrane pore channel (Fig. 4a, b). Notably, the transmembrane regions were distinctly discernible through the high-resolution density map (Supplementary Figs. 4 and 5). This finding is noteworthy because it is often challenging to achieve this resolution for analyzing PFTs, particularly in the transmembrane region. Detergents or lipids often interfere with the three-dimensional reconstitution of membrane proteins, leading to compromised resolution[25]. Apart from the transmembrane region, the cytolysin and β-trefoil lectin domains of assembled VcαHL were also clearly reconstructed in the cryo-EM map (Supplementary Fig. 5). In contrast, the β-prism lectin domain was not visualized in the map due to its flexible position in the complex, as indicated by the fuzzy cloud observed in 2D classification images (Fig. 4c, d). Given the successful reconstitution of the β-prism lectin domain within the liposome-integrated VCC structure[25], it appears plausible that this domain, exhibiting high flexibility, would be stabilized via its interaction with the membranes.

When compared to pro-VcαHL, the protomer of assembled VcαHL displays several conformational changes post pore formation, in addition to the substantial relocation of pre-stem loops (Fig. 4e). These include a slight clockwise rotation of the β-trefoil lectin domain when observed from the extracellular view and a minor displacement of the cradle loop, which seems to facilitate interaction with the adjacent protomer and release of the pre-stem loop (Fig. 4f). We propose that these minor structural changes could be attributed to the robust inter-subunit interactions that occur among the oligomerized protomers (Supplementary Fig. 6a). The β-trefoil lectin forms multiple intermolecular hydrogen bonds with neighboring β-trefoil lectin

domains (Supplementary Fig. 6b). Additionally, the cradle loop, cytolysin domain, and stem loop region all contribute to the inter-subunit interactions following the assembly of VcαHL into a ring-shaped complex (Supplementary Fig. 6c–e). Interestingly, despite VcαHL and VCC displaying distinct conformations during their immature state, they form remarkably similar structures post-assembly, as evidenced by a Cα R.M.S.D. of 0.38 Å (Supplementary Fig. 7a, b). The β-trefoil lectin domain in both VcαHL and VCC align to a similar position post pore formation (Supplementary Fig. 7c), suggesting that this assembled conformation represents an energy-favorable state stabilized by robust subunit interactions.

## Identification of the calcium binding module in assembled VcαHL elucidates the calcium stimulation mechanism

To elucidate the mechanism of calcium stimulation in VcαHL, we further attempted to find calcium ion binding sites in these VcαHL structures. In the cryo-EM map of assembled VcαHL, an extra-density in contact with residue H137 and E185 within the cytolysin domain was observed (Fig. 5a). In pro-VcαHL, H137 forms a hydrogen bond with a sulfate ion that binds to R313 of the pre-stem loop, while E185 participates in interactions with Q311 on the pre-stem loop (Fig. 5b). This sulfate ion forms a hydrogen bond network with water molecules, connecting E95 of pro domain and N183 of cytolysin domain in pro-VcαHL. These observations prompted us to hypothesize that the extra-density detected in the assembled VcαHL may be attributable to calcium ions. To further validate this hypothesis, we performed mutagenesis on residues H137, N183 and E185, which surround this extra-density, to elucidate their potential functional significance in VcαHL. Interestingly, these mutants exhibited approximately 40% hemolytic

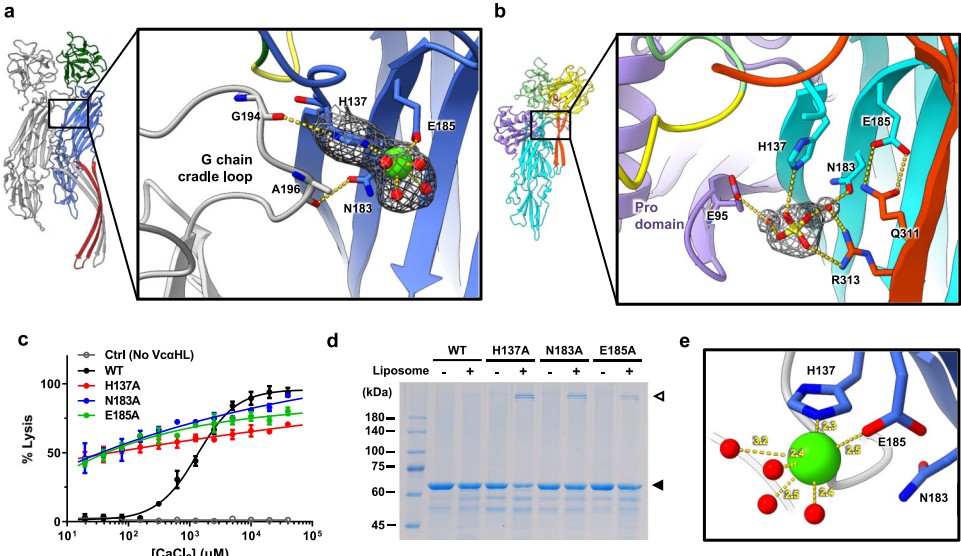

**Fig. 5 | Mutations in Ca$^{2+}$-binding residues impair the calcium-stimulating effect of VcαHL. a** The Ca$^{2+}$-binding sites in the assembled VcαHL are shown with the density map of the hydrated calcium ion and H137 in mesh. The Ca$^{2+}$ is presented as a green sphere and the residues involved in binding Ca$^{2+}$ are shown as sticks and colored by CPK. **b** The Ca$^{2+}$-binding residues in the pro-VcαHL structure interact with a sulfate ion, forming a hydrogen bonding network with adjacent waters and pre-stem loops. Yellow dashed lines indicate hydrogen bonds between molecules. The density map of the sulfate ion and waters (2Fo−Fc map contoured at 1.0 σ) were shown in mesh. **c** The hemolytic activities of the Ca$^{2+}$-binding residue mutants of

VcαHL were measured in the presence of various Ca$^{2+}$ concentrations. The VcαHL concentration is 7.5 µM. Each point represents the mean of three repeats, with error bars indicating the S.D. **d** SDS-PAGE analysis showed the Ca$^{2+}$-binding residue mutants of VcαHL could assemble on liposomal membranes in the absence of Ca$^{2+}$. Filled and open triangles indicate the unassembled and assembled VcαHL, respectively. **e** The calcium ion (green sphere) coordinates with H137, E185 and water molecules (red balls). The yellow dashed lines indicate the coordination bonds with labeled bond length.

activity even without calcium ions, suggesting a loss of their calcium-dependency (Fig. 5c). Moreover, oligomerization assays revealed that these mutants could partially assemble into oligomers without calcium ions (Fig. 5d). Negative stain TEM analysis further confirmed that these mutants formed ring-shaped complexes on the membrane in the absence of calcium ions, while the wild-type VcαHL could not (Supplementary Fig. 8). These findings lend support to our identification of a potential calcium binding site in VcαHL. We thus attributed this extra-density to a calcium ion, coordinated with four water molecules, in line with the size of the density and the observed bond distance of 2.3–2.5 Å (Fig. 5e), typical features of a hydrated calcium ion[26]. The hydrated calcium ions bind with H137 and E185, orienting the H137 to form a hydrogen bond with G194 from the cradle loop of an adjacent subunit. This suggests that Ca$^{2+}$ binding could enhance intermolecular interactions during VcαHL oligomerization. In addition, E185 in pro-VcαHL forms two hydrogen bonds with Q311 to maintain the association of the pre-stem loop with the cytolysin domain (Fig. 5b). These hydrogen bonds are replaced by coordination bonds with Ca$^{2+}$ in assembled VcαHL, potentially facilitating the detachment of the pre-stem loop and triggering subsequent pore formation processes (Fig. 5a). These structural details underscore the importance of these residues in calcium ion stimulation of VcαHL, thereby providing deeper insights into the role of calcium-dependent activation of this PFT.

### Histidine residues in VcαHL's rim region play essential roles in membrane association and toxin assembly

The cryo-EM map of VcαHL indicates the presence of densities, likely originating from SMA nanodiscs, around its transmembrane regions (Fig. 6a). These densities could delineate the transmembrane boundary of VcαHL[25,27], suggesting that the rim region penetrates into the membrane bilayer. Given that prior research highlights the importance of the membrane-embedded rim region for VCC's membrane-binding process[25,28], we further examined this region in VcαHL. Three histidine residues, H415, H422, and H426, were found to be clustered at the rim region below the membrane boundary (Fig. 6a). To explore the role of

this histidine cluster, we engineered alanine substitutions at these positions and assessed the resultant functional impacts on VcαHL.

Substantial decreases in hemolytic activity for the H415A and H422A mutants were observed, with a moderate decrease for H426A (Fig. 6b). The oligomerization assay confirmed that H415A and H422A were unable to form SDS-resistant complexes in the presence of calcium ions, indicating the critical contribution of these histidine residues to the toxin assembly process (Fig. 6c). Moreover, negative stain TEM analysis revealed that the H415A and H422A mutants retained their ability to form heptameric oligomers on the cholesterol-containing membranes in the presence of 10 mM CaCl$_2$, yet the rate of assembly was significantly curtailed in comparison to the wild type (Fig. 6d). The H415A and H422A substitutions resulted in fewer ring-shaped complexes, and many of these formed incomplete ring structures, suggesting that their assembly efficiency was reduced. In addition, ELISA assays further revealed lower membrane association affinity for H415A and H422A compared to H426A and the wild-type VcαHL (Fig. 6e, Supplementary Table 4). Together, these findings highlight the critical role of these membrane-embedded histidine residues in VcαHL's rim region for effective membrane interactions.

### High-resolution cryo-EM map of assembled VcαHL revealed a cation-π pair at the narrowest pore channel region

In addition, the cryo-EM map of the assembled VcαHL uncovers a distinctive feature within the pore channel—an unassigned extra density located between the side chains of R279 and W314 (Fig. 7a). The observed bond distance between the extra density and the W314 side chain is 3.4 Å, suggesting the possibility of a cation-π interaction (Fig. 7b)[29,30]. Considering the buffer composition used during cryo-EM grid preparation contained 150 mM potassium acetate, we propose the cation to be a K$^+$ ion. Subsequent real space refinement yielded an average correlation coefficient (CC) value of 0.85 for the seven potassium ions, supporting this structural interpretation. These K$^+$ ions are positioned at the narrowest section of the pore channel, reducing the pore radius to 4.16 Å (Fig. 7c, d). R279 and the K$^+$ ions

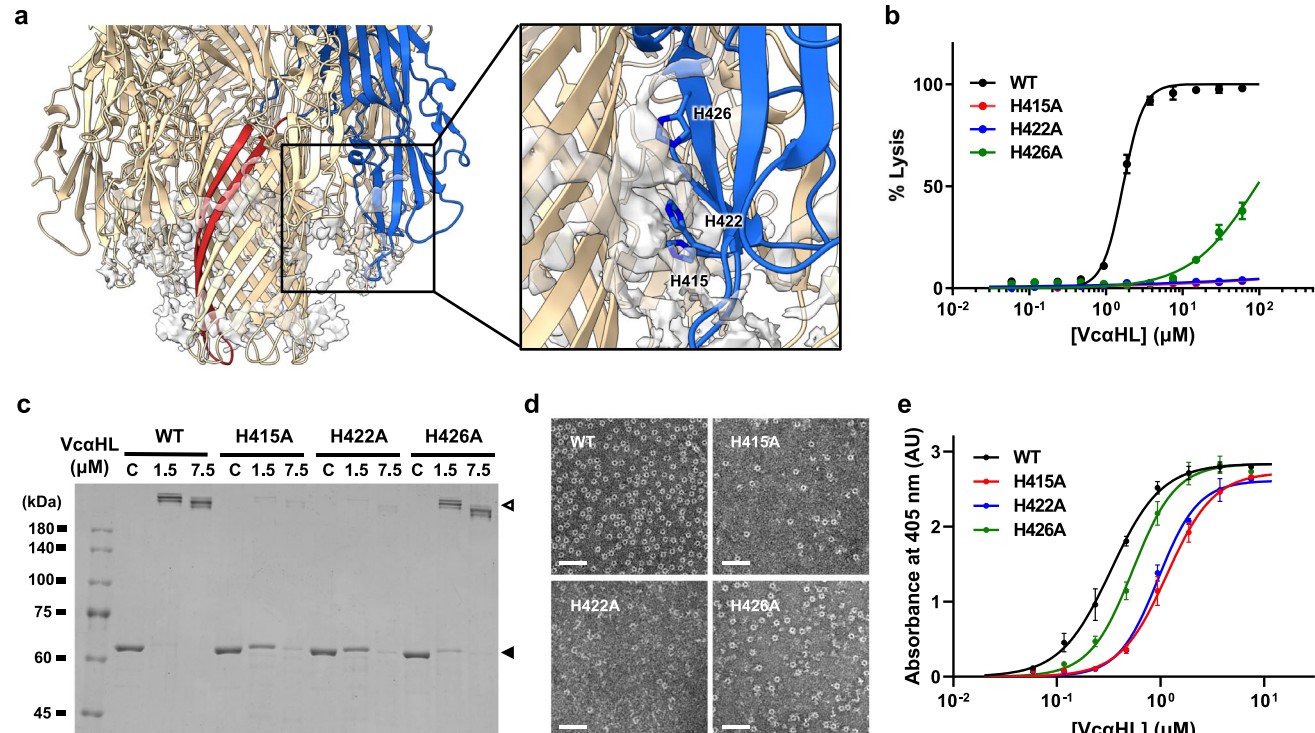

**Fig. 6 | The histidine residues in the rim region of VcαHL are critical for membrane binding. a** The coordinate model of VcαHL is displayed in cartoon, with a single protomer colored by domain. The extra-density around the transmembrane pore is rendered semi-transparent. An enlarged view of the boxed area shows the three histidine residues (H415, H422, and H426) clustered at the rim region. **b** Hemolytic activities of the H415A, H422A, and H426A mutants are compared to the wild type (WT). Each data point denotes the average from three independent experiments. Dose-dependent stimulation curves were generated using a nonlinear regression function, and the error bar represents the S.D. **c** The toxin assembly of these mutants was examined using SDS-PAGE. Both wild-type and mutant proteins were treated with liposomal membranes at protein concentrations of 1.5 and 7.5 μM, respectively. The sample C indicates the control group which is

the sample at protein concentrations of 1.5 μM without the addition of liposomal membranes. Three independent experiments yielded similar results. **d** Negative stain TEM shows that fewer complexes assembled on membranes in the H415A and H422A mutants compared to the wild-type and H426A. For each mutant, 10 independent micrographs yielded consistent results. The scale bar is 50 nm. **e** ELISA assay shows that H415A and H422A associated with immobilized membranes with lower affinities compared to the wild-type and H426A. The binding curves were fitted with the binding-saturation functions using non-linear regression. Each data point represents the mean of three independent samples ($n = 3$), with error bars representing the S.D. All assays were conducted in the presence of 10 mM CaCl$_2$. The liposomal membranes utilized in panels **c**–**e** contained 10% cholesterol.

create a positively charged region at the pore channel entrance, which may influence the channel's permeability (Fig. 7e). Interestingly, this potassium ion binding denotes a unique structural feature of VcαHL, absent in the previously characterized structure of VCC[12]. The structural superimposition of assembled VcαHL and VCC reveals a distinct side chain orientation of the tryptophan residue within the pore channel (Supplementary Fig. 7d). The distance between R279 and W314 expands in VcαHL to accommodate the potassium ion, whereas the K283 and W314 are closer in VCC. The distinctive positioning of W314 in the VcαHL structure presents an intriguing area for further investigation into channel flux and ion permeability, factors that might play a critical role in the pathogenic efficacy of this toxin.

## Discussion

The current study provides an in-depth analysis of the calcium-stimulation of VcαHL. By employing high-resolution structural characterization of both pro-VcαHL and assembled VcαHL, we identified the key residues and domains responsible for calcium stimulation, pore formation, and membrane association. These findings underpin a proposed working model that elucidates the calcium-stimulated pore formation mechanism in VcαHL (Fig. 8). In the proposed model, VcαHL is secreted in an inactive state, with its pre-stem loop bound to the cytolysin domain. Following proteolytic cleavage to remove the pro domain, calcium ions bind to residues H137 and E185, interrupting the interaction between pre-stem loop and cytolysin domain. This interruption triggers structural rearrangements, resulting in the release of

the pre-stem loop and causing the side chain of H137 to interact with the cradle loop from adjacent subunit. Concurrently, the β-trefoil lectin domain undergoes a subtle rotation to interact with neighboring subunits. Sequential oligomerization of VcαHL then forms a heptameric pre-pore on the membrane surface, and the liberated pre-stem loop constitutes a β-barrel transmembrane channel within cholesterol-laden lipid bilayers. This model provides insights into the ion-stimulation of PFTs, which could be beneficial for developing therapeutic strategies to control hemolytic attacks from bacterial pathogens.

Our findings provide insights into calcium-stimulated pore formation in PFTs. Calcium, a ubiquitous signaling ion, is pivotal to numerous intracellular and extracellular communication processes and responses[31]. Considering that the extracellular calcium concentrations can reach millimolar levels[23], which are comparable to the EC$_{50}$ for stimulating VcαHL's hemolytic activity (Fig. 1c), we propose that VcαHL may recognize host calcium as a trigger to initiate pore assembly upon its attachment to the host cell membrane. The requirement for high calcium levels and cholesterol content in membranes could ensure that VcαHL remains inactive until it reaches target cells, thereby enhancing the toxin's efficacy in targeting specific cells or tissues. It's worth noting that calcium also enhances the potency of several virulence factors secreted by *Vibrio* pathogens. For example, calcium can intensify bile salt-induced virulence activation in *V. cholerae* by controlling toxin-coregulated pilus[32]. Furthermore, calcium promotes pathogen adherence to host molecules, as

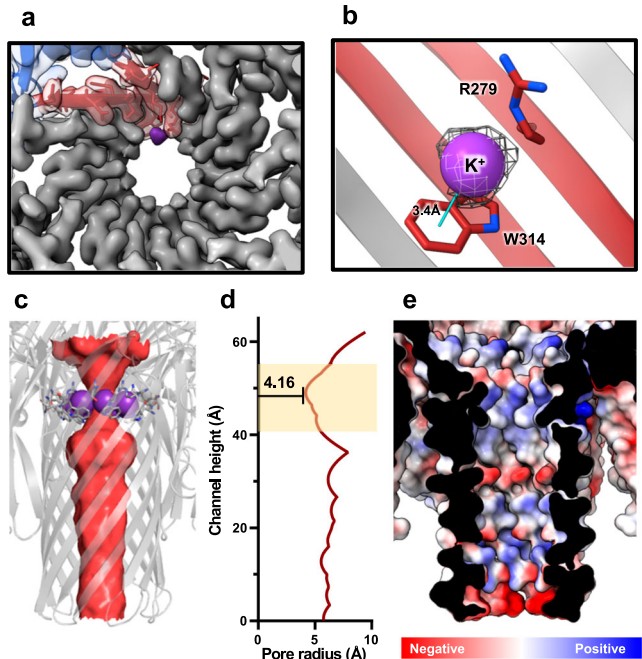

**Fig. 7 | The cryo-EM map reveals potassium ion binding sites at the narrowest section of the VcαHL pore channel. a** The cryo-EM density map of the transmembrane pore of VcαHL is shown, with a single protomer represented by different colors for each domain and displayed in a transparent manner to reveal the coordinates. The density map for the K⁺ ion is separated from the protein densities and marked in purple. **b** The K⁺ ion which is shown as purple sphere is bound to W314 through a cation-π interaction, shown as a cyan line. The gray mesh represents the EM density map of the K⁺ ion. **c** The transmembrane pore region of VcαHL is displayed in semi-transparent cartoon. The channel surface (colored in red) is displayed by using HOLLOW programs. The K⁺ ions (purple spheres), W314, and R279 constrict the channel radius. W314 and R279 are shown as sticks. **d** The channel radius is calculated by using HOLE software to reveal the radius profile along the channel height. The pore channel of VcαHL has a minimum radius of 4.16 Å. Yellow area indicates the pore region constricted by K⁺ ions, W314, and R279. **e** The electrostatic potential map of VcαHL's pore channel is shown to illustrate the charge distribution within the channel.

demonstrated by *V. vulnificus* requiring calcium for its attachment to chitin[33]. This capability to sense host calcium levels could provide a strategic advantage for pathogens, potentially increasing their virulence and survival. Like other calcium-regulated virulence factors, the calcium-dependent activation of VcαHL could potentially intensify host-pathogen interactions.

Furthermore, our structural findings illustrate that $Ca^{2+}$ binding is essential for dictating the conformational changes of VcαHL. The calcium-binding residues, H137, N183 and E185, are conserved across several *Vibrio* and *Aeromonas* species (Supplementary Fig. 1). Interestingly, mutations at these calcium-binding residues did not nullify the oligomerization and hemolytic activity of VcαHL. Instead, VcαHL could be activated without presence of calcium ions when its calcium-binding ability was impaired, suggesting that H137, N183 and E185 constitute a 'switch module' that suppress toxin oligomerization until calcium binding occurs. Residues H137 and N183 are implicated in binding a sulfate ion that links the pre-stem loop, pro-domain, and cytolysin domain in pro-VcαHL. E185, on the other hand, engages in direct hydrogen bonding with the pre-stem loop. Hence, the switch module likely acts to retain the pre-stem loop, thereby inhibiting toxin activation under conditions of low calcium concentration. When calcium levels rise, this switch module is activated via calcium binding and subsequent release of the pre-stem loop. In addition, this switch module exhibits ion-selectivity primarily for calcium ions,

although $Ba^{2+}$ and $Mn^{2+}$ can also mildly stimulate VcαHLs' hemolytic activity, whereas $Mg^{2+}$ cannot. The inability of $Mg^{2+}$ to stimulate VcαHL might be attributed to its shorter coordination bond distance. The theoretical coordination distance of $Mg^{2+}$-O is approximately 2.1 Å, while the $Ca^{2+}$-O is 2.4–2.5 Å[34,35]. The ideal coordination distance is essential for reorienting H137 and disrupting interactions between E185 and the pre-stem loop. Meanwhile, the bond distances of $Mn^{2+}$-O and $Ba^{2+}$-O are around 2.2 Å and 2.8 Å, respectively[26,34]. Therefore, $Mn^{2+}$ and $Ba^{2+}$ might still form partial interactions with H137 and E185, leading to a mild enhancement of VcαHL activity. Nonetheless, considering the increased hemolytic activity of the H137A and E185 mutants at higher calcium ion concentrations, the presence of additional calcium regulatory sites cannot be ruled out (Fig. 5c).

In addition, our study reveals that three histidine residues, H415, H422, and H426, clustered at the rim region of VcαHL, participate in membrane association (Fig. 6). Previous studies noted the complete integration of histidine residues H419 and H426 within the rim region of VCC into the lipid layer, and mutations at these sites have been linked to significant reductions in the hemolytic activities of VCC[18,25]. Despite these discoveries, the functional contribution of these rim-region histidine residues remains under-explored. Histidine residues are known mediators of protein-lipid interactions in several membrane proteins, particularly at the lipid-water interface[36,37]. Owing to the amphipathic nature of their imidazole rings under physiological conditions, residues H415 and H422 of VcαHL could peripherally interact with both the polar heads and nonpolar acyl chains of lipid bilayers, anchoring themselves at the membrane interface. Hypothetically, these histidine residues may facilitate the insertion of the rim region into membranes by inducing localized structural perturbations and disrupting lipid molecule packing. This theory is supported by the presence of histidine residues in the rim region of various PFTs, including *Staphylococcus aureus* α-hemolysin[38], γ-hemolysin[39], and *Clostridium perfringens* NetB toxin[40]. The inherent amphipathic characteristics of *Vibrio* αHLs may enhance their integration into the lipid bilayer of cell membranes, leading to non-specific associations with various target membranes[10,41]. Consequently, these non-specific membrane interactions potentially enable the pore formation of VcαHL.

Negative stain TEM images of VcαHL oligomerization highlight both ring-shaped heptameric oligomers and arc-shaped intermediates, offering visual evidence for the pore-forming process (Fig. 2). These arc-shaped intermediates, ranging from trimeric to heptameric formations, suggest that VcαHL likely undergoes a stepwise assembly process starting from a monomeric state. Yet, the specific environment in which this calcium-stimulated assembly occurs - in solution or on the membrane surface - remains uncertain. Our native PAGE analysis, which failed to detect VcαHL oligomers stimulated by calcium ions in the absence of membranes (Fig. 2c, lane 2), suggests that VcαHL may not form water-soluble oligomers in an aqueous environment. Instead, the assembly process appears to predominantly occur on the membrane surface. Consequently, we propose that these arc-shaped oligomers are intermediates participating in the oligomerization process and are not disintegrated fragments from the assembled water-soluble heptamers. This leads us to the question of whether these arc structures are intermediates attached to the membrane surface or if they have already integrated into the membranes. Membrane insertion pathways for β-PFTs can be divided into two categories: concerted and non-concerted[6,42]. Concerted β-PFTs form pre-pore complexes on membrane surfaces before membrane insertion, whereas non-concerted toxins could integrate into the membrane as monomers to form partial rings within the membrane. It is yet to be determined if VcαHL belongs to the category of concerted β-PFTs. However, the ability of VcαHL to form oligomers on cholesterol-free membranes without further assembly into SDS-stable heptamers provides a unique

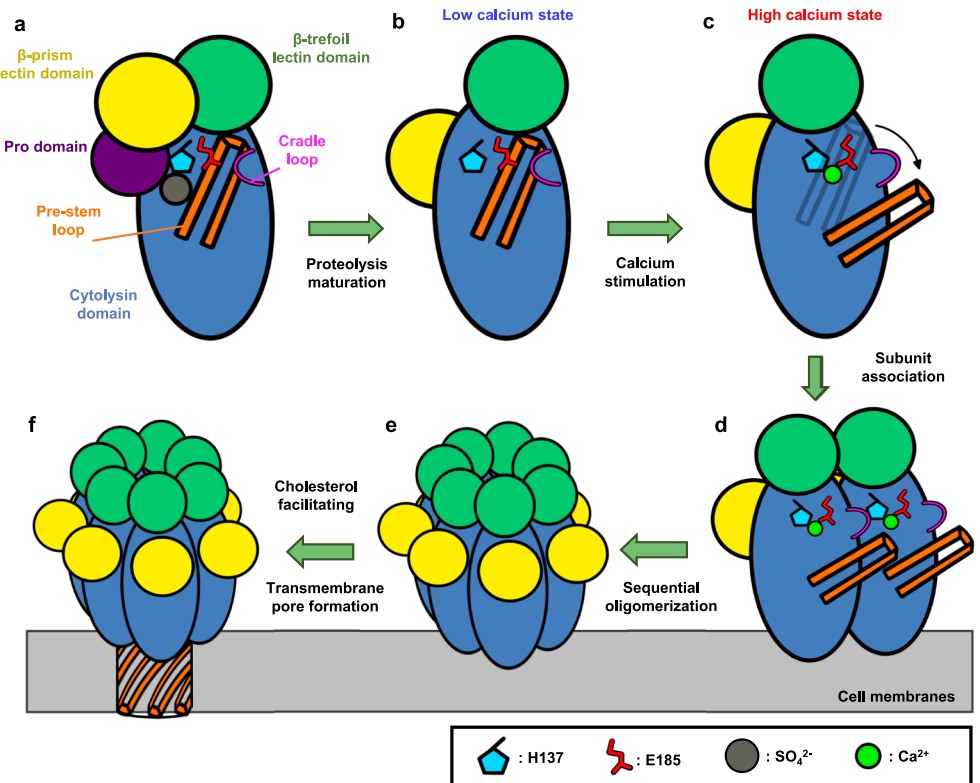

**Fig. 8 | A proposed mechanism of Ca²⁺-stimulated pore formation by VcαHL on the membrane. a** The pro-VcαHL is initially secreted from pathogens as an inactivated state, with the pro-domain bound to its cytolysin domain. H137 binds a sulfate ion, connecting the pre-stem loop, pro-domain, and cytolysin domain. E185 and the cradle loop engage with the pre-stem loop to maintain its contact with the cytolysin domain. **b** Upon proteolytic maturation, the pro-domain is cleaved, freeing the cytolysin domain to form inter-subunit interactions and waiting for oligomerization signals. **c** In high-calcium concentration environments, Ca²⁺ binding to H137 and E185 disrupts the interactions between the pre-stem loop and cytolysin domain, triggering a series of conformational changes in the cradle loop and pre-stem loop. **d** The association of calcium-bound subunits is initiated on the cell membrane. The cradle loop contacts the H137 residues of adjacent subunits, and the β-trefoil lectin domain also undergoes a slight rotation to contact the neighboring subunits. **e** Following sequential oligomerization, a pre-pore heptameric structure is formed on the membrane surface. **f** The stem loops penetrate the membrane and rearrange to form a β-barrel transmembrane pore. The presence of cholesterol in the membrane could facilitate this membrane integration and pore formation process.

opportunity to explore the membrane insertion process of *Vibrio* αHLs further. More detailed structural elucidation of these VcαHL intermediates could offer critical insights into the oligomerization processes and transmembrane pore formation.

The high-resolution cryo-EM map provided detailed insights into the VcαHL pore structure, identifying the cation bound to W314 at the narrowest region of the pore channel. The aromatic residues forming a 'φ-clamp' at the pore channel's entrance were previously observed in VCC and anthrax PA channel[43–45]. Previous studies indicate that replacing tryptophan with serine in VCC intensifies hemolytic toxicity, suggesting that the φ-clamp may modulate ion flux in *Vibrio* αHLs[46]. Our cryo-EM data further suggested that VcαHL's φ-clamp could bind to K⁺ or other cations, potentially regulating ion permeability through the pore channel. As VCC exhibits anionic preference for channel flux[47], the cation bound to the φ-clamp might assist in stabilizing anions within the pore and governing ion flow through the channel, thereby adjusting ion permeability and specific ion binding of these *Vibrio* αHLs.

In summary, this study uncovers the critical role of calcium ions in stimulating the hemolytic virulence factor of a *Vibrio* pathogen and identifies the molecular mechanism of toxin assembly triggered by these ions. These findings not only advance our understanding of the structure-function relationship inherent in *Vibrio* αHLs, but also pave the way for the development of targeted therapeutic strategies against infections caused by *Vibrio* species and related pathogens.

## Methods

### Plasmids construction

The plasmids used in this study were constructed to express the recombinant pro-VcαHL with a C-terminal His-tag, as described below. The gene encoding pro-VcαHL, without the N-terminal signal peptide, was amplified by PCR from the genomic DNA of *Vibrio campebellii* 12909. Subsequently, the amplified gene was inserted into the pET24a vector, between the *Nde*I and *Xho*I restriction cutting sites, to form the plasmid, pET-VcαHL. Plasmids for expressing point mutants were constructed using In-Fusion HD Cloning Kits (Cat. #638910, Takara Bio), with pET-VcαHL serving as the template. Primer sequences utilized to construct these expression plasmids were detailed in Supplementary Table 5.

### Protein expression and purification

The *E. coli* Rosetta-gami 2 strain, optimal for pro-VcαHL expression and ensuring correct disulfide bond formation, was utilized in our study. *E. coli* cells transformed with pET-VcαHL were initially grown overnight and then diluted in fresh LB medium. Protein expression was induced by adding 0.5 mM IPTG at the mid-log phase, maintained at 18 °C. Following 18 h of induction, the *E. coli* culture was centrifuged at 5000 × *g* for 10 min, and the harvested cells were washed with PBS to remove any residual medium. These cells were resuspended in lysis buffer (20 mM Tris-HCl, pH 7.6, 400 mM NaCl, 10 mM imidazole, and 1 mM phenylmethanesulfonylfluoride (PMSF)), and lysed by ultrasonication on ice. Cell debris was removed by centrifugation at

13,000 × $g$ for 30 min. The supernatant, containing the target proteins, was applied to a 5 mL His-trap column (PN 17524801, Cytiva) at a constant flow rate of 3 mL/min. After a 15-column volume (CV) wash with purification buffer (20 mM Tris-HCl, pH 7.6, 400 mM NaCl) containing 15 mM imidazole, protein elution was conducted using a stepwise imidazole gradient. Notably, pro-VcαHL primarily eluted under the 75 mM and 100 mM imidazole steps. The fractions containing highly pure target proteins were concentrated by using 10 kDa MWCO centrifugal filters (Amicon Ultra-15 mL, UFC901024, Millipore) and dialyzed in storage buffer (20 mM Tris-HCl, pH 7.6, and 150 mM NaCl).

## Proteolytic maturation of VcαHL

The purified pro-VcαHL was subjected to limited trypsin digestion to remove the pro-domain. This was performed using a calculated ratio of 11 trypsin units to 1 nmole of VcαHL. The mixture underwent incubation on a Rotamixer at 25 °C for 60 min. Post digestion, the proteins were concentrated by using 10 kDa MWCO centrifugal filters (Amicon Ultra-15 mL, UFC901024, Millipore) and then dialyzed using PD10 desalting columns (PN 17085101, Cytiva). They were then purified via a 1 mL HisTrap column (PN 17524701, Cytiva) to eliminate the cleaved pro-domain. The resulting mature VcαHL was eluted and concentrated to achieve a final protein concentration of 5 mg/mL. Finally, it was dialyzed in a storage buffer (20 mM Tris-HCl, pH 7.6, and 150 mM NaCl) in preparation for subsequent experiments.

## Liposome preparation

The liposome preparation and VcαHL oligomerization methods were referred to in previous studies with minor modifications[11]. We mixed 50 mg of phosphatidylcholine (PC) dissolved in 5 mL of chloroform and 5 mg of cholesterol dissolved in 1 mL of ethanol. The mixture was then fully dried in a 100 mL round-bottom flask under a stream of nitrogen gas. The dried lipid membrane was subsequently dissolved in 5 mL of liposome buffer (20 mM Tris-HCl, pH 7.6, and 150 mM NaCl), then extruded through a 0.1 μm filter 21 times. The prepared liposomes were used immediately for protein reconstitution.

## Toxin assembly of VcαHL

The liposome reconstitution method was based on previous studies with minor modifications[11]. Specifically, 5 mg of extruded liposomes were mixed with 0.5 mg of mature VcαHL and 10 mM CaCl$_2$ in the liposome buffer (20 mM Tris-HCl, pH 7.6, and 150 mM NaCl) and incubated at 25 °C for 30 min. Liposomes incorporated with VcαHL were collected by ultracentrifugation at 100,000 × $g$ for 30 min and then resuspended in the same liposome buffer. The oligomeric VcαHL on the liposome membrane could be detected in 10% SDS-PAGE without boiling the samples.

## Hemolysis assay

The hemolysis assay was adapted from a previous study with some modifications, as described below[18]. Defibrinated sheep blood, purchased from Taiwan Prepared Media (Cat. #B-TBS01000-100), was centrifuged at 400 × $g$ for 5 min to collect erythrocytes. The collected erythrocytes were washed 4 times with assay medium (20 mM Tris-HCl, pH 7.2, 150 mM NaCl, and 5% (w/v) glucose), and then resuspended to achieve a cell density of 3 × 10$^8$ cells/mL. Mature VcαHL was mixed with erythrocytes at a concentration of 1.5 × 10$^8$ cells/mL and 10 mM CaCl$_2$, and then incubated at 25 °C for 60 min. After removing cell debris by centrifugation at 800 × $g$ for 5 min, the absorbance of the supernatant was measured using a xMark™ microplate spectrophotometer (BIO-RAD) at 410 nm. The absorbance of each sample was normalized to the absorbance of a control sample treated with protein buffer as 0%, whereas treated with 1% (v/v) Triton X-100, which was designated as 100% lysis.

## Calcein release assay

The calcein release assay was adapted from a previous study with minor modifications[48]. The dried lipid membrane containing 50 mg PC and 5 mg cholesterol was dissolved in 5 mL of calcein assay buffer (20 mM HEPES, pH 8.0, and 150 mM NaCl) containing 50 mM calcein and was extruded ten times through a 0.1 μm filter. Free calcein was removed using Sephacryl™ S-300 desalting resin (PN 17059910, Cytiva), previously equilibrated with the calcein assay buffer. The eluted liposome solution was collected and further diluted to 10 mL using the calcein assay buffer, supplemented with CaCl$_2$ to adjust the final Ca$^{2+}$ concentration to 20 mM. Following this, 50 μL of the liposome solution was dispensed into a black microplate and mixed with an equivalent volume of mature VcαHL, the concentration of which varied from 0.12 to 30 μM. The calcein fluorescence was subsequently measured using a microplate fluorescence spectrometer (FLUOstar Omega, BMG Labtech) at an emission wavelength of 515 nm upon excitation at 488 nm. The calcein release was calculated as a percentage, where the fluorescence intensity of the negative control (with no added VcαHL) was considered as 0% and the intensity of the positive control (treated with 1% v/v Triton X-100) represented 100% release.

## Negative stain TEM analysis

The liposomal membranes incorporated with assembled VcαHL were freshly prepared and immediately placed on TEM grids (400 mesh, Formvar/Carbon Cu grid, Ted Pella) for 60 s. Excess samples were removed with filter paper, then the grids were washed twice with distilled water and treated with a droplet of 2% (w/v) uranyl acetate for another 60 s. The TEM grids were then allowed to fully air-dry before TEM image acquisition. Micrographs were collected using a TEM microscope (JEM-1400, JEOL). The accelerating voltage was set at 100 kV, and the pixel size was 2.5 Å/pix. TEM micrographs were processed with EMAN2[49] and SPHIRE[50] software for reference-free 2D classification.

## Blue native PAGE analysis

Blue Native PAGE (BN-PAGE) analysis was employed to study the oligomeric states of VcαHL, modifying the protocol from a previous study[51]. Mature VcαHL was assembled on liposomal membranes with or without cholesterol as described earlier. The assembled products were then solubilized in the liposome buffer (20 mM Tris-HCl, pH 7.6, and 150 mM NaCl) containing 1% β-OG at 4 °C for 15 min. The solubilized VcαHL samples were combined with an equal volume of 2X native sample buffer (50 mM Imidazole, 500 mM 6-Aminocaproic acid, 0.1% Coomassie blue G-250 and 10% (w/v) glycerol). For the BN-PAGE, we prepared an 8% uniform acrylamide running gel and a 3.5% stacking gel (acrylamide/bis-acrylamide, 19:1). The samples were run at 180 V at 4 °C for 3 h using a cathode buffer (50 mM Tricine/HCl, pH 7.0, 7.5 mM Imidazole, 0.002% Coomassie blue G-250) and an anode buffer (25 mM Imidazole/HCl, pH 7.0). The gel was subsequently stained with 0.1% Coomassie Blue R250 in a solution composed of 10% acetic acid, 50% methanol, and 40% H$_2$O to enhance visibility.

## Crystallization, X-ray data collection and structure determination

The purified pro-VcαHL was concentrated to 40 mg/mL for crystallization. Initial conditions were obtained from the Wizard Classic 2 crystal screen kit, specifically condition Tube 12 (100 mM sodium cacodylate/HCl, pH 6.5, 30% (v/v) PEG 400, 200 mM lithium sulfate). After optimization, pro-VcαHL protein crystals were grown over three days at 20 °C using hanging-drop crystallization experiments against a reservoir buffer (100 mM sodium cacodylate/HCl, pH 6.0, 200 mM lithium sulfate, and 30% (v/v) PEG 400). X-ray diffraction experiments were conducted using synchrotron radiation with a wavelength of 1.00 Å at the TLS BL13B of NSRRC in Taiwan. Diffraction images were

collected with Bluice software, and the data were processed and scaled using HKL2000. The phase was determined by molecular replacement using the initial model predicted by Alphafold2 (AFID: A0A344KRS4) as the template. Model refinement was performed using the PHENIX suite[52], and manual modeling was conducted between refinement cycles using COOT[53]. Data collection and refinement statistics are provided in Supplementary Table 2. All structural figures were generated using ChimeraX[54].

### Preparation of VcαHL-SMALPs

The assembled VcαHL in liposomal membranes was extracted using SMA2000, with minor modifications to a previous study's protocol[55]. Liposomes integrated with 2 mg of assembled VcαHL were collected by ultracentrifugation at $100,000 \times g$ for 30 min. The liposomes were then resuspended in the liposome buffer (20 mM Tris-HCl, pH 7.6, and 150 mM NaCl) containing 2.5% (w/v) SMA and incubated at 25 °C for 2 h. After another round of ultracentrifugation at $100,000 \times g$ for 30 min, the solubilized VcαHL-SMALPs remained in the supernatant fraction while insolubilized liposomes were removed. The VcαHL-SMALPs were concentrated to a final protein concentration of 4 mg/mL and applied to a size exclusion chromatography (SEC) column (Superose 6 Increase 10/300 GL, Cytiva) at a constant flow rate of 0.5 mL/min with SEC buffer (25 mM HEPES-KOH, pH 7.6, and 150 mM potassium acetate). The peak fractions were collected for subsequent cryo-EM analysis.

### Cryo-EM grids preparation

Cryo-samples were prepared using a Vitrobot Mark IV (Thermo Fisher Scientific) with a temperature setting of 4 °C and a humidity of 100%. An aliquot (~4 μL) of purified VcαHL-SMALPs was applied to the glow-discharged Quantifoil R1.2/1.3 holey carbon grid (Quatifoil GmbH, Germany). After a 10 s wait, the grids were blotted with filter paper for 3.5 s and quickly plunged into liquid nitrogen pre-cooled liquid ethane. After vitrification, the cryo-EM grids are stored in liquid nitrogen until imaging.

### Cryo-EM data acquisition

The cryo-EM data set is automatically collected by EPU-2.7.0 software (Thermo Fisher Scientific) on a 300 kV Titan Krios (Thermo Fisher Scientific) equipped with X-FEG electron source. The data were collected by a K3 Summit detector (with GIF Bio-Quantum Energy Filters, Gatan) operating in super-resolution mode. The raw movie stacks were recorded at a nominal magnification of 81,000×, corresponding to a pixel size of 1.061 Å/pixel (super resolution 0.5305 Å/pixel). The defocus range was set to −1.5 to −2.25 μm and the slit width of Energy Filters was set to 20 eV. Forty frames of non-gain normalized tiff stacks were recorded with a dose rate of ~23 $e^-/Å^2$ per second and the total exposure time was set to 2.15 s, resulting in an accumulated dose of ~50 $e^-/Å^2$ (~1.25 $e^-/Å^2$ per frame).

### Single-particle image processing and 3D reconstruction

Image stacks of VcαHL-SMALPs underwent motion-correction and dose-weighting using MotionCor2[56], followed by the calculation of the contrast transfer function (CTF) using CTFFIND4.1[57]. Particle picking, extraction, and classification were carried out in cisTEM[58] and Relion[59]. In the final round of 2D classification, the particles in good 2D classes were transferred to cryoSPARC[60] for ab initio map generation without imposing any symmetry (C1). The ab initio map was then imported into Relion[59] as a starting reference for further 3D auto-refine with C7 symmetry. Shiny particles were transferred to cryoSPARC[60] for additional 2D classification and 3D refinement, with the final resolution estimated using Fourier Shell Correlation (FSC) = 0.143. The coordinate model was constructed based on the SWISS-MODEL predicted homolog model (template model structure: PDB 3O44[12]) and refined with the Phenix software suite[52] and COOT[53]. Cryo-EM reconstructions

and model refinement details are presented in Supplementary Figs. 3, 4 and Supplementary Table 3. Visualization of 3D density maps, coordinate models, pore radius, and channel shapes was achieved using UCSF ChimeraX[54], HOLLOW[61] and HOLE[62].

### ELISA

A modified ELISA from a previous study[48] was used to detect VcαHL's association with the liposome. Microplates were coated with 100 μg of PC and 10 μg of cholesterol dissolved in methanol, and the solvent was evaporated overnight under vacuum. Wells were then rinsed thrice with PBS and blocked with 5% (w/v) milk in PBS for 1 h. VcαHL at various concentrations was added and incubated for 1 h at 25 °C with 10 mM $CaCl_2$ to aid membrane association. Bound VcαHL was detected using a mouse anti-his primary antibody (Anti-Histidine Tagged Antibody, clone HIS.H8, Cat. #05-949, Lot. 3660392, 1:8000 dilution, Millipore) and a goat anti-mouse IgG secondary antibody conjugated with alkaline phosphatase (PN NEF824001EA, Lot. 120015, 1:10000 dilution, PerkinElmer). After three washes with PBS, a 100 μL substrate solution (4.5 mM pNPP, 1 M diethanolamine, 0.5 mM MgCl2, pH 9.8) was added to each well and incubated for 30 min at 25 °C. Absorbance was measured at 405 nm with a xMark™ microplate reader (BIO-RAD). The absorbance values of the samples were plotted against VcαHL concentration using GraphPad Prism, and the $K_d$ value was calculated by performing non-linear regression fitting to the binding saturation function.

### Reporting summary

Further information on research design is available in the Nature Portfolio Reporting Summary linked to this article.

## Data availability

The data that support this study are available from the corresponding author upon request. The cryo-EM density map has been deposited in the Electron Microscopy Data Bank (EMDB) under accession code, EMD-36150 (assembled VcαHL). The atomic coordinates have been deposited in the Protein Data Bank (PDB) under accession codes, 8JBQ (pro-VcαHL) and 8JC7 (assembled VcαHL). The initial model used to determine the crystal structure of pro-VcαHL for molecular replacement was obtained from the Alphafold2 database, AFID: A0A344KRS4. The model coordinates of VCC used for structural comparison were obtained from PDB, 1XEZ (pro-VCC) and 3O44 (assembled VCC). Source data are provided with this paper.

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

## Acknowledgements

We gratefully acknowledge the experimental facility and the technical services provided by the Synchrotron Radiation Protein Crystallography Core Facility of the National Core Facility for Biopharmaceuticals, Ministry of Science and Technology, and the National Synchrotron Radiation Research Center (NSRRC), a national user facility funded by the Ministry of Science and Technology, Taiwan (R.O.C.). Initial crystallization screening was carried out at the NSRRC-NCKU Protein Crystallography Laboratory at University Center for Bioscience and Biotechnology of National Cheng Kung University (funding by NSRRC 11123LAB02). The cryo-EM experiments were performed at the Academia Sinica Cryo-EM Center (ASCEM) and the cryo-EM data were processed at the Academia Sinica Grid-computing Center (ASGC). ASCEM is supported by Academia Sinica (Grant number AS-CFII-108-110) and Taiwan Protein Project. ASGC is supported by Academia Sinica. We also appreciate the use of TEM instruments (EM000900 of MOST 110-2731-M-006-001) belonging to the Core Facility Center of National Cheng Kung University. This work was supported by funding from Academia Sinica (AS-TP-112-L01) and Taiwan Protein Project (AS-KPQ-109-TPP2) to M.C.H. and the Ministry of Science and Technology (MOST) of Taiwan (MOST 109-2636-B-006 –001; MOST 110-2636-B-006 –012; MOST 111-2636-B-006 –009; NSTC 112-2311-B-006 –007) to S.M.L.

## Author contributions

S.M.L. conceived and initiated the project; S.M.L., Y.C.C., Y.A.C., and H.C. designed and carried out biochemical and mutagenesis experiments. M.C.Y., C.H.W., M.C.H., and S.M.L. designed and performed cryo-EM analysis and structural determination. S.M.L., Y.C.C., and Y.A.C. conducted X-ray diffraction experiments, model building and structural refinements. H.Y.L., Y.A.C., and S.M.L. cloned the gene and constructed the DNA plasmids for protein expression. All authors analyzed the experimental results and prepared and edited the figures and tables. S.M.L., Y.C.C., M.C.Y., and M.C.H. wrote and edited the manuscript.

## Competing interests

The authors declare no competing interests.
