## [Peer Review File · Nature Communications]

Structural basis for calcium-stimulating pore formation of
Vibrio α -hemolysinReviewers' Comments:

Reviewer #1:

Remarks to the Author:

The manuscript by Chiu et al describes an in depth functional and structural characterisation of Vibrio alpha-hemolysin protein (VcaHL). VcaHL is related to several characterise pathogenic pore-forming proteins including Vibrio cholerae cytolysin which are critical for invasion into host tissue.

It is noteworthy, and commendable, that the authors have produced the highest resolution pore-state of an alpha-hemolysin protein by using SMA in their cryo-EM preparation. This includes resolving features such as calcium binding and a hypothesised potassium binding site in the lumen of the beta-barrel. In the case of calcium binding, this is the central to the proposed molecular mechanism of VcaHL whereby Chiu et al. describe a role in preventing untimely membrane binding, oligomerisation/pore-insertion. Furthermore, the authors obtained the X-ray structure of VcaHL representing the precursor form of the protein (prior to oligomerisation). This state also revealed a possible mechanism of inhibition by coordination with a sulfate by several residues. The authors have attempted to validate most of the structural findings by producing mutant variants to test the effect on calcium binding and pore-formation. In summary, the authors have nicely described the molecular mechanism of VcaHL with respect to the role of calcium in mediating pore-formation.

Regardless, there are a some issues I would like to see addressed to improve the manuscript.

There are a few issues with the nomenclature throughout the manuscript that the authors might not be aware of. For example, in the pore-forming field, typically, 'pre-pore' is reserved for describing an oligomeric intermediate that is not inserted. Throughout the manuscript the authors have often used pre-pore to mean the precursor form of the protein. I suggest for consistency with this field that they change this throughout (see specific comments below). Perhaps they can adopt pro-VcaHL or simply precursor.

Line 74 - It would be helpful for context if the authors could provide analysis in Supp 1 to show the domain identity between VcaHL and VCC. This will help the reader identify which regions are most conserved / variable. Further, the alignment in Supp 1 would be stronger if the domains of these proteins were shown rather than just the secondary structure.

Line 90 - the authors dont explain the implication of this result. I.e. what does it mean that they form a high molecular weight complex resistant to SDS in this context. This may be obvious to experts in pore forming biology but not to the general audience

Line 94 - In panel 1B, it is not clear what is meant by 'mock'. It is left to the reader to believe this may be no calcium. It seems that an obvious experiment to be included in this panel would be a haemolytic assay containing EDTA to see if the activity would be reduced compared to no calcium and with calcium treatment. Did the authors test this?

Line 96 - A measurements between the EC 50 is not performed in any statistical manner. So it is not clear what is meant by significantly lower.

Line 98 - While the authors claim about the physiological concentration of calcium could be correct, it is unclear if there is calcium pre-loaded in the protein that was scavenged during the purification process. If the authors added chealating agents (e.g. EDTA) during the purification, it would be worth mentioning in the results to clarify this.

Line 103 - as per the previous comment it is not obvious to the reader what is meant by 'mock' in this context. Presumably various concentrations of protein in a buffer that does not contain calcium. It would be better stated simply as 'no calcium'

Line 103 - It would appear that in panel E the authors have used a single protein concentration to probe the effect of various calcium concentrations. However, the protein concentration is not provided in the figure nor the legend. It is also not stated how this concentration for the assay was selected (i.e. what is the rationale of this concentration?).

Line 120 - It seems that there is a bit of inconsistency between the results displayed in 2A and 2B. The negative stain sample of VcaHL with calcium but no cholesterol seems to form oligomers to the same degree as the sample containing cholesterol. I suggest, that this may even represent two separate and discrete assemblies. Maybe helpful for the authors for future study?

Line 129 - This seems presumptuous for the authors to call these pre-pore oligomers in lieu of a description of the structural evidence.

Line 135 - As commented previously, this seems premature to call this a pre-pore.

Line 138 - The authors may be confusing the terminology of 'pre-pore' with precursor in the context of pore assembly in this context.

Line 141 - Presumably, this is with respect to pro-VCC. If that is the case, then it should be stated

Line 152 - Perhaps this is a mistake, but it seems as though panel 3D shows the interaction between the B-trefoil and the B-prism domain, not the pre-stem loop interactions. It would actually, be helpful for clarity to the reader if the pre-stem loop was coloured differently so that it stands out from the B-prism domain.

Line 157 - I believe this is meant to be Fig 3E?

Line 159 - Fig 3F?

Line 176 - Did the authors attempt to perform any localised refinement in an attempt to resolve this domain?

Line 182 - Stated this way makes it seem as though Fig 4D is meant to be a superposition between the precursor and the pore state. This does not seem to be the case, or perhaps this is not clear. Perhaps the authors can change the figure to make this change stand out better to the reader.

Line 186 - It is not clear what is meant by 'instigated' in this context. Are the authors suggesting that the interactions between oligomerised subunits are what triggers pore formation / conformational change?

Line 199 - while it is plausible to propose this is indeed a calcium binding site, it would be more convincing with traditional experimental techniques. Do the authors observe calcium binding in their X-ray structure?

Line 219 - Given that these residues are all involved in coordinating to the sulfate in the pro-VcaHL form, is it not more likely that these mutations have lost this interaction and as such produced a non-auto-inhibited form of the proteins?

Line 223 - As previous, the authors may confuse readers that study pore forming proteins by using pre-pore in this context. Further, it is not clear why these are not simply called pores? Are they not membrane inserting?

Line 236 - It would be more appropriate for the authors to do a comparison based on EC50 values

(similar to assay in Fig 1B). The analysis as displayed in 6B, while significant, only represents a single concentration and may lead the reader to believe H415A and H422A are completely inactive. Indeed the authors own results (negative stain and membrane binding ELISA) demonstrate this is not the case. It is perfectly valid, in my view, to state relative to wild-type how the EC50 differ with these mutants.

Line 254 - as someone who is unfamiliar with this type of interaction, is it not surprising to the authors that the potassium would be located between the W and R given the charge on potassium?

Line 265 - It is unclear if the authors are suggesting that this is a consequence of the resolution they have achieved or if is a unique difference between VcaHL and VCC

Line 283 - this is probably the only correct usage of pre-pore

Line 351 - This is actually a very dated paper for PA structure. I suggest this should be updated with one of higher resolution.

Line 388 - This would be more meaningful to the reader at a mol:mol ratio, rather than w/w.

Line 390 - MWCO not mentioned?

Line 402 - Minor point. This seems odd that the authors would choose an even number of extrusions as it suggests they collect their sample from the side that would contain the excluded membrane material. From my understanding most would collect from the opposite side from the initial extrusion.

Line 421 - Authors should also designate what the 0% normalized control was. i.e. was this buffer only or cells treated with no protein?

Line 430 - Perhaps there is a detail missing from the previous sentence. It is not understood how the authors know what the liposome concentration is after performing desalting step? How was this measured?

Line 448 - It would be helpful for the reader if the condition was stated (XX buffer, salt, precipitant)

Fig 8 - For clarity to the reader, it might be helpful if some of the features could stand out better. For example, it is a bit unclear what H137 and E185 are doing in this schematic. Especially as these components are quite small in the figure.

Reviewer #2:

Remarks to the Author:

Chiu et al describe the structure of Vibrio alpha hemolysin soluble monomer and heptameric pore. They report calcium, potassium, and sulfate ions as well as lipids in their maps. Based on these observations they designed point mutations and assessed hemolysin function. They show that calcium is necessary for the formation of SDS-resistant pores and for pores to reach their highest lytic activity. While this work undoubtedly contributes to our understanding of Vibrio alpha hemolysin, I have several comments and suggestions for the authors to improve clarity and accuracy.

1. Figure S1. The authors could label the different domains such as lectin and cytolysin for ease of understanding.
2. Have the authors attempted to solubilize in SMA the non-SDS resistant oligomers that are formed in the presence of calcium but absence of cholesterol and subsequently solved their structure? This

would strengthen the sequential model presented in figure 8. In addition a native PAGE would further support the claim of SDS-sensitive oligomers.

3. Interpretation of rings and arcs: The paper could benefit from a discussion on whether the arcs are intermediate stages of ring formation, or if they correspond to rings that fell apart. Do they have any cryo-EM images of arcs?

4. The β -prism lectin domain is expected to bind the receptor. It is located quite far away from the plasma membrane. Can the author speculate on candidate receptors taking this piece of information into account?

5. The authors mention a cradle loop but it is not quite clear to me what it is in the structure. Can they highlight it in figure 4? Why does it have a special name? Is it so critical for the function of the protein? These points should be explained in the text so that a non pore-forming specialist reader can understand the subject.

6. In the PDB of the pore provided by the authors, 7 chains (H to N) constitute a second molecule. Each of these chains consists of 5 amino acid peptide. They are numbered Gly 298 to Lys 302. Their sequence is identical to the sequence of amino acids 298 to 302 in the first molecule, which consists of chains A to G. The scores for molecule B in the PDB validation report are extremely low. This makes me suspicious of a modeling error, and in particular of an error done by Modelangelo. In the methods (lines 497 to 499), the authors mention that the coordinate model was constructed based on the SWIS-MODEL predicted homologue model and refined with Phenix and COOT. What was the homologue model used? In addition, the loop does not roughly fit an extra density in the EM map. What is the nature of this density?

7. The reported resolution seems overestimated. This is based both on visual inspection and on the PDB validation report. At page 39 of the report (section 8.2 resolution report), the table shows that the resolution reported by the authors is 2.06 Angstrom, while the resolution calculated by the PDB validation website from the deposited half-maps is 2.76 Angstrom, which is good but significantly less high than reported. In my experience this is a common issue of cryoSPARC. It very often gives resolution estimates significantly better than Relion, while the maps output by both programs look very similar by visual inspection. Could it be that a too tight mask was used to estimate resolution? Could we see for the two half maps output by cryoSPARC a resolution estimated by Relion, or RESMAP, using their standard masking procedure?

The reported local resolution in the barrel is between 2.4 and 2.8. However most side chains are significantly less well resolved than that. It looks as if this region was rather above 3.0 A. In this respect, the sentences between lines 170 and 173 are misleading.

8. The paper mentions a 'slight clockwise rotation' on page 11, line 183. Could the authors specify the perspective for this observation? Side view? Cytoplasmic view? Extracellular view?

9. Page 12. The current narrative, where the authors attribute extra density to hydrated calcium, can lead to misinterpretations, especially given the presence of a high number of similar densities throughout the map. The authors should be more nuanced. Yet the suggestion of the calcium binding site arises from a convergence of indications, including the binding of a sulfate ion by H137 in the crystal structure and the existence of another density in contact with this residue in the EM map. This putative binding site is further substantiated by the loss of Ca²⁺ dependency following mutagenesis of the residues surrounding this density.

10. In alignment with the previous comment, the map's current clarity does not substantiate any assertions about the presence of lipid density. At the threshold used to see the density shown in fig 6, the protein is covered with extra densities, it looks like a noisy blob. It is impossible just based on the density to claim that a lipid is present there. The claimed lipid density resembles SMA molecules

surrounding the transmembrane region of beta barrels that have not been exposed to lipids. A more balanced wording would be that other hemolysins were shown to interact with lipids at the rim domain. This was confirmed for VcaHL by the presence of a His cluster in the rim domain which, when mutated, lead to loss of binding. The legend of Fig. 6 should also mention if calcium was added and in what concentration. Regarding the statement on page 14, line 245, about fewer pre-pore complexes resulting from substitutions, it would be interesting to have a native PAGE here as well (see point 2).

11. Following the previous point, the discussion at pages 19 and 20 about lipid binding should be made less bold. While the quality of the map is overall good, claiming that the TM resolution is so high and reveals discernible lipids is an overstatement.

12. Materials and methods. More details would be helpful regarding:

-plasmid construction - a table with primers, possibly a plasmid map.

- protein expression - the range of concentrations used in the stepwise imidazole gradient elution should be indicated.

13. Have the authors tried to improve the resolution of the beta barrel by masking out or subtracting the fuzzy densities of SMA density and of the beta-prism lectin domain?

Point by point response to reviewer's comments:

Reviewer #1 (Remarks to the Author):

The manuscript by Chiu et al describes an in depth functional and structural characterisation of Vibrio alpha-hemolysin protein (VcaHL). VcaHL is related to several characterise pathogenic pore-forming proteins including Vibrio cholerae cytolysin which are critical for invasion into host tissue.

It is noteworthy, and commendable, that the authors have produced the highest resolution pore-state of an alpha-hemolysin protein by using SMA in their cryo-EM preparation. This includes resolving features such as calcium binding and a hypothesised potassium binding site in the lumen of the beta-barrel. In the case of calcium binding, this is the central to the proposed molecular mechanism of VcaHL whereby Chiu et al. describe a role in preventing untimely membrane binding, oligomerisation/pore-insertion. Furthermore, the authors obtained the X-ray structure of VcaHL representing the precursor form of the protein (prior to oligomerisation). This state also revealed a possible mechanism of inhibition by coordination with a sulfate by several residues. The authors have attempted to validate most of the structural findings by producing mutant variants to test the effect on calcium binding and pore-formation. In summary, the authors have nicely described the molecular mechanism of VcaHL with respect to the role of calcium in mediating pore-formation.

Regardless, there are a some issues I would like to see addressed to improve the manuscript.

There are a few issues with the nomenclature throughout the manuscript that the authors might not be aware of. For example, in the pore-forming field, typically, 'pre-pore' is reserved for describing an oligomeric intermediate that is not inserted. Throughout the manuscript the authors have often used pre-pore to mean the precursor form of the protein. I suggest for consistency with this field that they change this throughout (see specific comments below). Perhaps they can adopt pro-VcaHL or simply precursor.

Response: We apologize for any misunderstanding. Our use of the term 'pre-pore' was meant to describe the oligomeric intermediate state of VcaHL that is assembled on the membrane surface but has yet to integrate into the membrane, which aligns with the conventional definition in the pore-forming toxin field. We appreciate your concern about the consistency of our terminology. To clarify our description and avoid potential confusion, we've revised our manuscript to ensure we use the term 'pre-pore' strictly in the context of describing this specific intermediate state of VcaHL. Thank you for pointing out this important aspect, and we are confident this adjustment will contribute to the clarity and accuracy of our manuscript.

Line 74 - It would be helpful for context if the authors could provide analysis in Supp 1 to show the

domain identity between V α HL and VCC. This will help the reader identify which regions are most conserved / variable. Further, the alignment in Supp 1 would be stronger if the domains of these proteins were shown rather than just the secondary structure.

Response: We appreciate this suggestion. We have now included an analysis in Supplementary Table S1 that shows the domain identity between V α HL and VCC. This new analysis highlights the conserved and variable regions between these proteins. Additionally, we have enhanced the alignment by including the domains of these proteins alongside the secondary structure, as you recommended. We believe this additional information will provide valuable context to the readers.

Line 90 - the authors don't explain the implication of this result. I.e. what does it mean that they form a high molecular weight complex resistant to SDS in this context. This may be obvious to experts in pore forming biology but not to the general audience.

Response: You are correct that this point was not sufficiently explained for a general audience. We have now added a clarification in the manuscript at Page 6, Line 92. We explain that the formation of a high molecular weight complex resistant to SDS indicates that the proteins are forming stable pore structure, which is characteristic of pore-forming proteins during the process of inserting into membranes. This information should make the manuscript more accessible to readers who are not experts in the field.

Line 94 - In panel 1B, it is not clear what is meant by 'mock'. It is left to the reader to believe this may be no calcium. It seems that an obvious experiment to be included in this panel would be a haemolytic assay containing EDTA to see if the activity would be reduced compared to no calcium and with calcium treatment. Did the authors test this?

Response: We apologize for any confusion caused by the use of the term 'mock'. We have clarified in the manuscript that 'mock' refers to a control experiment without calcium. Additionally, we appreciate your suggestion regarding the inclusion of a hemolytic assay with EDTA. We have now performed this experiment and included the results in panel 1B. This new data shows the effect of chelating calcium with EDTA on the hemolytic activity of V α HL, providing additional insights into the role of calcium.

Line 96 - A measurements between the EC 50 is not performed in any statistical manner. So it is not clear what is meant by significantly lower.

Response: We appreciate your comment on the need for statistical clarity in the presentation of our EC₅₀ data. In response, we have now provided the 95% confidence intervals (CI) for the EC₅₀ values in the manuscript. When comparing the EC₅₀ for calcium-stimulated hemolytic activity with the control group lacking calcium ions, the 95% confidence intervals of the two groups do not overlap. This lack of overlap implies statistically significant differences. We thought this approach provides a more robust and statistically appropriate representation of our findings.

Line 98 - While the authors claim about the physiological concentration of calcium could be correct, it is unclear if there is calcium pre-loaded in the protein that was scavenged during the purification process. If the authors added chelating agents (e.g. EDTA) during the purification, it would be worth mentioning in the results to clarify this.

Response: We appreciate your insightful query. To clarify, we did not incorporate chelating agents during the protein purification process to eliminate any pre-loaded calcium in V α HL. We are confident that pre-loaded calcium should not significantly influence the EC₅₀ of calcium ions on V α HL activity due to the following reasons:

1. Presently, there is no existing evidence to suggest that pro-V α HL can bind calcium ions without the removal of the pro domain. In the crystal structure of pro-V α HL, calcium binding residues associate with the pre-stem loop and a sulfate ion. Consequently, calcium ions may not pre-load in the pro-V α HL during purification.
2. The concentration of calcium inside *E. coli* cells is approximately 90 ± 10 nM (*J Biol Chem.* (1987) 262, 12570), which is markedly lower than the EC₅₀ of calcium ions stimulating V α HL. Prior to cell lysis, the *E. coli* cells were washed with PBS without calcium ions. The buffers used in purification were also devoid of calcium ions. Hence, the concentration of environmental calcium ions during protein preparation would be considerably below the millimolar level, insufficient to notably affect the EC₅₀ determination.
3. If pro-V α HL naturally accumulates calcium ions during purification, it suggests that the native pro-V α HL secreted by *V. campbellii* would also carry a baseline level of these ions. Upon entering host tissues, this could trigger hemolysis. Therefore, the experimentally calculated EC₅₀ of calcium likely provides an accurate indication of the calcium ion requirements for physiological functions.

Given these reasons, we believe that the current measurement of the EC₅₀ sufficiently substantiates our claim concerning the physiological concentration of calcium.

Line 103 - as per the previous comment it is not obvious to the reader what is meant by 'mock' in this context. Presumably various concentrations of protein in a buffer that does not contain calcium. It would be better stated simply as 'no calcium'

Response: We appreciate your feedback. You're correct, the term 'mock' may not be immediately clear to all readers. To improve clarity, we have replaced 'mock' with 'No CaCl₂' throughout the manuscript. Similarly, the control group with no added proteins is now denoted as 'Ctrl (No V α HL)'. These changes have been reflected in the figure legends as well. Thank you for helping us enhance the clarity of our manuscript.

Line 103 - It would appear that in panel E the authors have used a single protein concentration to probe the effect of various calcium concentrations. However, the protein concentration is not provided in the

figure nor the legend. It is also not stated how this concentration for the assay was selected (i.e. what is the rationale of this concentration?).

Response: Thank you for pointing this out. We have now included the protein concentration used in panel E in the figure legend. Additionally, we have provided a rationale for the selection of this concentration in the manuscript at Page7, Line103.

Line 120 - It seems that there is a bit of inconsistency between the results displayed in 2A and 2B. The negative stain sample of VcaHL with calcium but no cholesterol seems to form oligomers to the same degree as the sample containing cholesterol. I suggest, that this may even represent two separate and discreet assemblies. Maybe helpful for the authors for future study?

Response: Thank you for your insightful comments. You're correct to observe some discrepancy between the results in Figures 2A and 2B, which reveal both the roles of calcium and cholesterol in VcaHL pore formation. As you pointed out, the ring-shaped oligomers depicted in Figs. 2A and 2B likely represent two distinct assembly states with different resistance to SDS-denaturation. Prior research suggests that cholesterol may physically interact with *Vibrio* α HLs during pore formation, possibly aiding the development of β -barrel transmembrane pores in the membrane. (*Microbiol Immunol.* (2006), 50, 751) Consequently, we propose that the ring-shaped oligomers formed on cholesterol-free membranes likely constitute a pre-pore structure without integrating into the membrane, therefore, sensitive to SDS denaturation.

In our revised manuscript, we've included additional blue native PAGE analysis (Fig. 2C), showing that the oligomers assembled on cholesterol-free membranes would disassemble upon membrane solubilization with the gentle detergent, β -OG. This finding supports our hypothesis that these oligomers could be 'pre-pores', residing on the membrane surface prior to insertion. Lacking full membrane integration, these ring-shaped oligomers are unable to sustain their complex structure, leading to their reversion to monomers after solubilization.

We remain grateful for your valuable input, which has not only spurred further examination but also helped improve the clarity of our manuscript.

Line 129 - This seems presumptuous for the authors to call these pre-pore oligomers in lieu of a description of the structural evidence.

Response: Thank you for your keen observations. Indeed, we have included a blue native PAGE analysis in our revised manuscript, aiming to better elucidate the nature of these ring-shaped complexes that are stimulated by the presence of calcium ions. We've observed that these ring-shaped heptamers, formed on cholesterol-free membranes, would substantially dissociate into monomers following membrane solubilization. In addition, no intermediate oligomer states (3-6 mers) observed in either SDS-PAGE or blue native PAGE, suggesting that these intermediate states attached on membrane

surface without forming the stable transmembrane β -barrels. Consequently, we have proposed these SDS-sensitive, ring-shaped oligomers as potential pre-pore complexes of V α HL. Therefore, we proposed that calcium ions stimulate the pre-pore formation stage during the pore-forming processes of V α HL. Recognizing, however, that the original description may have introduced some confusion, we have carefully revised our language to describe these oligomers more accurately. The revised paragraphs is at Page 9, Line 142. The revised manuscript should provide a more precise discussion of the roles that cholesterol and calcium ions play in the pore-forming processes of V α HL.

Line 135 - As commented previously, this seems premature to call this a pre-pore.

Response: In line with your previous comment, we have revised our language to avoid the premature designation of these complexes as 'pre-pores'. We believe that our updated description provides a more accurate interpretation of our experimental findings.

Line 138 - The authors may be confusing the terminology of 'pre-pore' with precursor in the context of pore assembly in this context.

Response: We appreciate your attention to detail. In our previous responses, we indicated that we consider the heptameric ring structures formed on cholesterol-free membranes as pre-pore complexes of V α HL, given their propensity to dissociate into monomers after membrane solubilization. These ring-shaped oligomers, comprised of seven protomers, can be stimulated in the presence of calcium ions (Fig. 2b). As such, we suggest that calcium ions catalyze the sequential oligomerization of V α HL, facilitating pre-pore formation on the membrane surface. When cholesterol is present in the membrane, these pre-pore oligomers may further develop into transmembrane pore complexes capable of resisting SDS-denaturation and membrane solubilization, thanks to their stable β -barrel structure. Consequently, we posit that calcium ions play a crucial role in the pre-pore formation stage of V α HL's pore-forming processes. Nevertheless, to improve clarity and avoid confusion, we have made careful adjustments to our manuscript's terminology, replacing the term 'pre-pore' with 'ring-shaped oligomers'. This modification should more accurately represent our hypothesis.

Line 141 - Presumably, this is with respect to pro-VCC. If that is the case, then it should be stated

Response: You are correct that the structural comparison we referred to was indeed with respect to the pro-VCC. We have now explicitly stated this in the revised sentence at Page 10, Line 168.

Line 152 - Perhaps this is a mistake, but it seems as though panel 3D shows the interaction between the B-trefoil and the B-prism domain, not the pre-stem loop interactions. It would actually, be helpful for clarity to the reader if the pre-stem loop was coloured differently so that it stands out from the B-prism domain.

Response: You are correct. Panel 3d shows the interaction between the β -trefoil and the β -prism domain, not the pre-stem loop interactions. To enhance clarity for our readers, we've made modifications to the figure and now have distinctly colored the pre-stem loop.

Line 157, 159 - I believe this is meant to be Fig 3E? Fig 3F?

Response: We appreciate your diligence. You're correct in observing that the references to the figures were misplaced. We have made the necessary corrections in the manuscript to accurately align the figure references.

Line 176 - Did the authors attempt to perform any localised refinement in an attempt to resolve this domain?

Response: In this study, we did not pursue further localized refinement to resolve the β -prism lectin domain individually. This decision was based on prior research showing that the β -prism lectin acts as a ridge domain in VCC, and the individual β -prism lectin domain has been clearly elucidated in the crystal structure of pro-V α HL. In the context of this research, gaining insights into the relative position of the β -prism lectin domain to the main body of V α HL was deemed more significant than understanding the individual β -prism lectin domain. Nonetheless, the β -prism lectin domain did not demonstrate a specific position relative to V α HL's main body after being solubilized as SMALPs, making it challenging to achieve high-resolution structure of the β -prism lectin domain in conjunction with the main body.

Therefore, in this work, we opted not to further discuss the location of the β -prism lectin domain. We believe that prior studies that have resolved the VCC structure on the liposome membrane using cryo-EM would provide more reliable insights into the functional role of the β -prism lectin domain (*J Cell Biol.* (2021), 220, e202102035). In future studies, we intend to resolve the membrane-bound V α HL, which should offer more information about the functional role of this unique domain.

Line 182 - Stated this way makes it seem as though Fig 4D is meant to be a superposition between the precursor and the pore state. This does not seem to be the case, or perhaps this is not clear. Perhaps the authors can change the figure to make this change stand out better to the reader.

Response: We appreciate the feedback. Fig. 4D indeed represents the superposition between the protomers of the premature and pore states. To prevent any confusion that may arise due to an excess of colors, we have revised the figure to display the pro-V α HL model without color. This alteration aims to accentuate the structural differences more clearly. We believe this modification will enhance the clarity of the figure and more effectively illustrate the conformational changes that occur post transmembrane pore formation.

Line 186 - It is not clear what is meant by 'instigated' in this context. Are the authors suggesting that

the interactions between oligomerised subunits are what triggers pore formation / conformational change?

Response: We appreciate the reviewer pointing out the ambiguity in our use of the term 'instigated.' In this context, we meant to suggest that the conformational changes may be the consequence by the robust subunit contacts after oligomerization. To provide greater clarity, we have revised the sentence to state: 'We propose that these relatively minor structural changes could be attributed to the robust inter-subunit interactions that occur among the oligomerized protomers' (Page 13, Line 216) This alteration should make our intentions clearer and avoid any potential confusion.

Line 199 - while it is plausible to propose this is indeed a calcium binding site, it would be more convincing with traditional experimental techniques. Do the authors observe calcium binding in their X-ray structure?

Response: We appreciate your suggestion. Indeed, we are also very interested in locating the calcium ions in the X-ray structure of pro-V α HL. However, no potential densities that could be annotated as calcium ions were observed in the map of pro-V α HL. Attempts to co-crystallize with calcium ions also did not yield any additional densities in the X-ray maps. As a result, we hypothesize that the pro-V α HL may not bind to calcium ions until the pro-domain is removed. Nevertheless, there are several indications supporting the calcium binding of mature V α HL, as evidenced by the hemolytic assay, biochemical data and structural findings. When we mutated the residues H137, N183 and E185, the loss of Ca²⁺ dependency further strengthened the validity of this putative binding site. We have now revised our manuscript to better illustrate the multiple lines of evidence for identifying the calcium binding site in the cryo-EM map. The revised descriptions, which can be found on Page13, Line 229, provide a more robust account of the calcium binding to these key residues.

Line 219 - Given that these residues are all involved in coordinating to the sulfate in the pro-V α HL form, is it not more likely that these mutations have lost this interaction and as such produced a non-auto-inhibited form of the proteins?

Response: You make a valid point. Our data indeed suggests that residues H137, N183, and E185 play a critical role in suppressing the oligomerization of the toxin prior to calcium binding. We hypothesize that these residues constitute a 'switch module' that controls the activation of V α HL. In the pro-V α HL form, both H137 and N183 are implicated in sulfate binding, whereas E185 appears to play a key role in connecting the pre-stem loop via hydrogen bonding. Hence, we hypothesize that these residues may be critical in maintaining the pre-stem loop in its appropriate position within the pro-V α HL form. Upon calcium binding, the pre-stem regions would be released from the switch module, and resulting in initiating toxin oligomerization.

Line 223 - As previous, the authors may confuse readers that study pore forming proteins by using pre-

pore in this context. Further, it is not clear why these are not simply called pores? Are they not membrane inserting?

Response: We appreciate your comment and apologize for any resulting confusion. We initially used the term 'pre-pore' because the negative stain images can only confirm the formation of ring-shaped complexes, but it cannot definitively prove membrane insertion of these V α HL complexes. Figures 2a and 2b illustrate that these ring-shaped complexes can form on cholesterol-free membranes without necessarily establishing stable beta-barrel pores. Therefore, we use pre-pore to emphasize calcium ions majorly functions in stimulating oligomerization but not membrane insertion of . However, we understand that using pre-pore to describe the oligomers on liposomal membrane is also not accurate. We have revised the sentence and avoid further misleading in current manuscripts.

Line 236 - It would be more appropriate for the authors to do a comparison based on EC₅₀ values (similar to assay in Fig 1B). The analysis as displayed in 6B, while significant, only represents a single concentration and may lead the reader to believe H415A and H422A are completely inactive. Indeed the authors own results (negative stain and membrane binding ELISA) demonstrate this is not the case. It is perfectly valid, in my view, to state relative to wild-type how the EC₅₀ differ with these mutants.
Response: We appreciate this suggestion. We have now performed an analysis based on EC₅₀ values and included this data as figure 6b in the manuscript. This provides a more comprehensive view of the activity of the mutants relative to the wild-type.

Line 254 - as someone who is unfamiliar with this type of interaction, is it not surprising to the authors that the potassium would be located between the W and R given the charge on potassium?

We acknowledge your perspective and understand why the location of the potassium ion between the tryptophan (W) and arginine (R) residues may seem surprising, given potassium's charge. We initially attempted to model Arg side chain directly interacting with Trp residues, but the experimental density map did not align with this interpretation. The size of the observed density could not be adequately accounted for by Arg side chains in the cryo-EM map. Thus, we proposed that the high potassium ion concentration in the protein buffer (exceeding 150 mM) might lead to the occupancy of K⁺ ions at this location, thereby accounting for the observed density in the map.

Line 265 - It is unclear if the authors are suggesting that this is a consequence of the resolution they have achieved or if is a unique difference between V α HL and VCC

Response: We apologize if our initial statement was unclear. To clarify, we believe the observation reflects a unique difference between V α HL and VCC, rather than a consequence of the resolution achieved. Notably, the crystal structure of the VCC complex, obtained at a comparable resolution of 2.9 Å, provided well-defined side chain positioning in the electron density map. Hence, we can confidently differentiate the orientation of the tryptophan residue's side chain (W314 in V α HL versus

W318 in VCC). This difference isn't a consequence of resolution limits but indicates a structural variation between V α HL and VCC.

Line 283 - this is probably the only correct usage of pre-pore

Response: Thank you for the affirmation. We have made sure to use the term 'pre-pore' consistently and accurately throughout the manuscript.

Line 351 - This is actually a very dated paper for PA structure. I suggest this should be updated with one of higher resolution.

Response: Thank you for pointing this out. We have updated the reference to a more recent paper with a higher resolution PA structure.

Line 388 - This would be more meaningful to the reader at a mol:mol ratio, rather than w/w.

Response: We appreciate your valuable suggestion. We acknowledge that the enzyme activity of trypsin can vary across different batches. Therefore, expressing the enzymatic activity in terms of unit-to-mole ratio, rather than by weight, would provide a more precise and meaningful understanding of our methodology. We have accordingly revised our manuscript to incorporate this trypsin unit-to-protein mole ratio, thereby enhancing the clarity and comprehensibility of our study.

Line 390 - MWCO not mentioned?

Response: We apologize for the oversight. We have now included the MWCO in the materials and methods of revised manuscript.

Line 402 - Minor point. This seems odd that the authors would choose an even number of extrusions as it suggests they collect their sample from the side that would contain the excluded membrane material. From my understanding most would collect from the opposite side from the initial extrusion.

Response: Thank you for your keen observation. We appreciate the opportunity to clarify this point. In our methodology, we consider one cycle of extrusion to consist of a forward and backward pass through the extruder, effectively mixing the sample. We collected the sample from the opposite side of the initial extrusion, which is in line with standard practice. To avoid any confusion, we have revised the manuscript to state that we performed **21 extrusions**, which accurately reflects the number of complete forward and backward passes. This clarification ensures that the methodology is transparent and accurately represented.

Line 421 - Authors should also designate what the 0% normalized control was. i.e. was this buffer only or cells treated with no protein?

Response: You are correct; this information is important for clarity. We have now specified in the manuscript that the 0% normalized control was cells treated with buffer only.

Line 430 - Perhaps there is a detail missing from the previous sentence. It is not understood how the authors know what the liposome concentration is after performing desalting step? How was this measured?

Response: Thank you for your insightful comment. We understand that our original description lacked clarity. We indeed determined the liposome concentration based on the initial lipid quantity used for liposome preparation. Since liposomes are approximately 100 nm in size, they are large enough to pass through the desalting column without retention, thus ensuring a high recovery rate. Thus, we initially estimated the liposome concentration considering the volume increase post-desalting. However, upon reflection, we realize that this might not provide the most accurate replication information for readers. Hence, in the revised manuscript, we now directly specify the final volume to which we diluted for these experiments. Corresponding adjustments have been made to clearly outline the steps we used to measure calcein release activity. We sincerely thank you for your insightful feedback, which has significantly contributed to enhancing the clarity and quality of our manuscript.

Line 448 - It would be helpful for the reader if the condition was stated (XX buffer, salt, precipitant)

Response: We appreciate your attention to detail. In order to enhance the clarity of the information, we have now explicitly provided the specific conditions of Tube 12 in the manuscript at Page 29, Line 522, including the type of buffer, salt concentration, and the precipitant used. We hope this additional information will help readers to better understand the experimental context.

Fig 8 - For clarity to the reader, it might be helpful if some of the features could stand out better. For example, it is a bit unclear what H137 and E185 are doing in this schematic. Especially as these components are quite small in the figure.

Response: We appreciate your suggestion for improving the clarity of Fig 8. We have updated the figure to ensure that key features, such as H137 and E185, are more prominent. We have also included annotations in the figure legend to elucidate their roles. To further enhance readability, we have increased the overall size of each figure. We believe these modifications will significantly enhance the clarity of Fig 8, facilitating a more comprehensive understanding of our research findings for the reader.

Reviewer #2 (Remarks to the Author):

Chiu et al describe the structure of Vibrio alpha hemolysin soluble monomer and heptameric pore. They report calcium, potassium, and sulfate ions as well as lipids in their maps. Based on these observations they designed point mutations and assessed hemolysin function. They show that calcium is necessary for the formation of SDS-resistant pores and for pores to reach their highest lytic activity. While this work undoubtedly contributes to our understanding of Vibrio alpha hemolysin, I have several comments and suggestions for the authors to improve clarity and accuracy.

1. Figure S1. The authors could label the different domains such as lectin and cytolysin for ease of understanding.

Responses: We appreciate this suggestion. We have now modified the Figure S1 to clearly label the distinct domains, including the pro-domain, cytolysin domain, pre-stem loop, β -trefoil lectin and β -prism lectin domain. We believe that these additions will help to improve the comprehensibility of the figure.

2. Have the authors attempted to solubilize in SMA the non-SDS resistant oligomers that are formed in the presence of calcium but absence of cholesterol and subsequently solved their structure? This would strengthen the sequential model presented in figure 8. In addition, a native PAGE would further support the claim of SDS-sensitive oligomers.

Response: Thank you for your insightful suggestions. We have indeed attempted to solubilize the non-SDS resistant oligomers from cholesterol-free liposomes using both detergents and SMA. Unfortunately, the oligomers seemed to dissociate after solubilization when analyzed using blue native PAGE, although some remaining oligomers could still be observed in comparison to the control. We've incorporated the blue native PAGE in the revised manuscript as Figure 2C to further elaborate on the crucial role of liposomal membranes in V α HL oligomerization. Negative-stain TEM consistently revealed oligomerization on the cholesterol-free liposome membrane. This leads us to hypothesize that membrane association is crucial to sustaining the oligomeric interaction. Support for this hypothesis comes from the absence of intermediate states (3-6 mers) in the blue native PAGE for the V α HL assembled on cholesterol-containing liposomes. The intermediate states formed on membrane surface could be clearly visualized in negative stain TEM but not observed after membrane solubilization. Therefore, we thought that the membrane association is a prerequisite for pre-pore formation of V α HL. Consequently, the intermediate states or pre-pore complex may not remain stable upon membrane solubilization.

3. Interpretation of rings and arcs: The paper could benefit from a discussion on whether the arcs are intermediate stages of ring formation, or if they correspond to rings that fell apart. Do they have any

cryo-EM images of arcs?

Response: We concur that this is a pivotal point to discuss. In the revised manuscript, we have now included the potential nature of rings and arcs, exploring whether the arcs could be intermediate stages of ring formation. These description could be found at Page 21, Line 378.

In regard to your query about the cryo-EM images of arcs, we regret to inform that we currently do not possess any such images. We share your interest in acquiring structural information of these intermediate states and the pre-pore structure. However, these arcs appear to exist only on the membrane surface, and membrane solubilization seems to cause significant disassembly of these intermediate states, making it challenging to obtain the soluble intermediate states for single particle analysis via cryo-EM.

We have attempted to assemble V α HL on an EM grid coated with a membrane bilayer. Unfortunately, the thick ice layer presents an obstacle in obtaining high-quality images for membrane-bound proteins. Previous studies suggest that directly visualizing these intermediates on low concentration liposomes might offer a solution. We plan to pursue this line of investigation in our future research and are optimistic about unraveling this intriguing question.

4. The β -prism lectin domain is expected to bind the receptor. It is located quite far away from the plasma membrane. Can the author speculate on candidate receptors taking this piece of information into account?

Responses:

Thank you for posing this thought-provoking question. While it is true that the β -prism lectin domain of V α HL is seemingly distant from the plasma membrane, several studies have shown that this domain can interact with membrane-associated carbohydrates that extend from the cell surface. Potential receptors could include a variety of glycoproteins or polysaccharides, which are often known targets of lectin domains. The spatial configuration of the β -prism lectin domain might be advantageous for binding to such protruding extracellular entities. Moreover, it could potentially interact with receptors that are part of larger protein complexes or have high molecular weights, which would typically reach further from the membrane surface.

Additionally, it is conceivable that the β -prism lectin domain serves to facilitate membrane surface association by binding to membrane receptors but dissociates after pre-pore oligomerization or pore formation. In this scenario, soluble V α HL might initially adhere to the membrane surface via interactions between the β -prism lectin domain and a membrane receptor. Subsequently, the rim region of V α HL could integrate into the lipid bilayer, thereby providing the energy necessary for dissociation of the β -prism lectin domain and membrane receptors.

However, these speculations are contingent upon further experimental validation, and there might be other viable mechanisms or receptor candidates to explore. Our current dataset does not provide definitive information on the exact position of the β -prism lectin domain in the assembled V α HL, thus we did not delve into a detailed discussion on the role of the β -prism lectin domain in this study. Nonetheless, your question provides valuable direction for our continued research into the intricate functionality of V α HL.

5. The authors mention a cradle loop but it is not quite clear to me what it is in the structure. Can they highlight it in figure 4? Why does it have a special name? Is it so critical for the function of the protein? These points should be explained in the text so that a non pore-forming specialist reader can understand the subject.

Responses:

We apologize for any confusion. The term ‘cradle loop’ was first introduced in previous studies to describe the structural feature that cradles the pre-stem loops in the water-soluble state of the protein (*Proc Natl Acad Sci U S A* (2011), 108(18), 7385). It is given this special name due to its important role in connecting neighboring subunits after pore formation, thus crucial to the toxin's function. To make this more clear to readers, we have highlighted the cradle loop with a different color in Figure 4 and have expanded on its structural role and significance within the protein's context in the text (Page 11, Line 183). We have also endeavored to make our explanation of why it is termed as the ‘cradle loop’ and its significance more accessible for readers who are not specialists in pore-forming proteins.

6. In the PDB of the pore provided by the authors, 7 chains (H to N) constitute a second molecule. Each of these chains consists of 5 amino acid peptide. They are numbered Gly 298 to Lys 302. Their sequence is identical to the sequence of amino acids 298 to 302 in the first molecule, which consists of chains A to G. The scores for molecule B in the PDB validation report are extremely low. This makes me suspicious of a modeling error, and in particular of an error done by Modelangelo. In the methods (lines 497 to 499), the authors mention that the coordinate model was constructed based on the SWISS-MODEL predicted homologue model and refined with Phenix and COOT. What was the homologue model used? In addition, the loop does roughly fit an extra density in the EM map. What is the nature of this density?

Responses:

We appreciate your keen observation. For our cryo-EM structure of V α HL, we constructed the coordinate model based on the homologous model predicted by SWISS-MODEL, which was based on the VCC structure (PDB: 3O44). Given that V α HL and VCC share 89% sequence identity and possess highly similar overall structures (see Supplementary Fig. S7), we were able to fit the predicted model into our cryo-EM map using Chimera, followed by further refinement with Phenix and Coot. We did not utilize automated modeling tools to build the coordinate model in this study. We have now clarified the information about the homologous template in the revised manuscript's materials and methods

section (Page 32, Line 572) to improve transparency.

We apologize for any confusion surrounding chains H to N. These chains represent fragments of the pre-stem loop that we manually built into the cryo-EM map using Coot. Upon analyzing the cryo-EM map post-coordinate model building, we noticed extra densities at the interface between the beta-trefoil lectin and cytolysin domains. Comparing with the crystal structure of pro-V α HL, these densities could fit the fragments from Gly 298 to Lys 302 of the pre-stem loop.

However, these densities were relatively weak, and only a few amino acids could be interpreted with our current maps. This observation suggests that some protein particles may retain their pre-stem loop without full release, given that the cryo-EM map was reconstructed using numerous particle images. We

attempted to separate these particles using 3D Var or 3D classification but were unable to isolate a distinct dataset with enhanced extra densities at this region, possibly due to limited particle numbers.

While these unreleased pre-stem loops might provide insights into the intermediate states of the pore-forming process, our current cryo-EM map does not offer solid evidence for a reliable interpretation or discussion about these fragments. Hence, we have removed the discussion about these fragments from this manuscript draft. We acknowledge that the scores for these fragments in the PDB validation report are indeed low, implying an unsuitable interpretation of the model. Consequently, we have removed chains H to N from the coordinate model of assembled V α HL and refined it further for PDB deposition. We have updated the modeling refinement information in Table 2 of the revised manuscript. In future studies, we aim to gather more structural data on these unreleased pre-stem loops by using cryo-EM to provide a detailed understanding of the pore-forming process of this fascinating toxin.

7. The reported resolution seems overestimated. This is based both on visual inspection and on the PDB validation report. At page 39 of the report (section 8.2 resolution report), the table shows that the resolution reported by the authors is 2.06 Angstrom, while the resolution calculated by the PDB validation website from the deposited half-maps is 2.76 Angstrom, which is good but significantly less high than reported. In my experience this is a common issue of cryoSPARC. It very often gives resolution estimates significantly better than Relion, while the maps output by both programs look very similar by visual inspection. Could it be that a too tight mask was used to estimate resolution? Could we see for the two half maps output by cryoSPARC a resolution estimated by relion, or

RESMAP, using their standard masking procedure?

The reported local resolution in the barrel is between 2.4 and 2.8. However most side chains are significantly less well resolved than that. It looks as if this region was rather above 3.0 Å.

In this respect, the sentences between lines 170 and 173 are misleading.

Responses:

Thank you for your valuable review. We appreciate your feedback on the potential variation in resolution due to different software and masks, despite using a gold standard FSC. In our study, cryoSPARC was used for the final reconstruction, and we reported the best resolution determined by cryoSPARC in the main text. We also included local resolution estimates in Supplementary Figure 4C to capture the resolution variations within the structure. In addition, we conducted a detailed analysis of resolution estimates using various masks (no mask, spherical, loose, and tight) in Supplementary Figure 4D. Here is a summary of the results:

We observed that the resolution reported by cryoSPARC without using a mask (2.8 Å) is similar to the resolution of 2.76 Å indicated by the PDB validation report for the unmasked structure. In response to your suggestion, we utilized cryoSPARC-generated half maps and mask, and imported them into Relion for resolution estimation. The obtained resolution using Relion was 2.18 Å, which falls within the range of resolutions reported by cryoSPARC using the Loose and Tight masks. Additionally, we performed local resolution estimation using cryoSPARC, Relion, and ResMap, as suggested. Please refer to the detailed results of these analyses below:

cryoSPARC (Supplementary Figure 4C)

Relion

ResMap

During the local resolution estimation, we noticed a small difference in the resolution estimates between Relion and cryoSPARC. Relion provided a slightly higher resolution estimate, approximately 0.1 Å higher than cryoSPARC. Additionally, we observed that ResMap tended to overestimate the local resolution compared to both cryoSPARC and Relion. It's important to take these variations into consideration when interpreting the local resolution results obtained from different algorithms, as you suggested.

Further, we have supplemented our findings with Supplementary Figure 5, which displays the alignment of the density map with the coordinate models at different regions. The figure notably highlights clear resolution in the side chain densities within the transmembrane (TM) region. We trust that this added data strengthens the reliability and robustness of the structure elucidated in our study.

8. The paper mentions a 'slight clockwise rotation' on page 11, line 183. Could the authors specify the perspective for this observation? Side view? Cytoplasmic view? Extracellular view?

Responses: We appreciate your careful attention to detail. Our description of a 'slight clockwise rotation' was indeed missing a clear perspective. We've updated the manuscript (Page 13, Line 214) to clarify that this rotation is observed from the extracellular viewpoint. Thank you for helping improve the clarity of our paper.

9. Page 12. The current narrative, where the authors attribute extra density to hydrated calcium, can lead to misinterpretations, especially given the presence of a high number of similar densities throughout the map. The authors should be more nuanced. Yet the suggestion of the calcium binding site arises from a convergence of indications, including the binding of a sulfate ion by H137 in the crystal structure and the existence of another density in contact with this residue in the EM map. This putative binding site is further substantiated by the loss of Ca^{2+} dependency following mutagenesis of the residues surrounding this density.

Responses: We appreciate your thoughtful comments and suggestions. Indeed, we concur that attributing the additional density directly to hydrated calcium ions could potentially lead to misinterpretations. To address this, we have meticulously revised our manuscript to present our observations in a more nuanced way as shown in Page 13, Line 229. We now underline that the identification of the calcium binding site is predicated on a convergence of several indicators. Further, we elaborate on the functional role of calcium binding, based on the structural evidence. We hope this amended presentation adequately addresses your concerns.

10. In alignment with the previous comment, the map's current clarity does not substantiate any assertions about the presence of lipid density. At the threshold used to see the density shown in fig 6, the protein is covered with extra densities, it looks like a noisy blob. It is impossible just based on the density to claim that a lipid is present there. The claimed lipid density resembles SMA molecules surrounding the transmembrane region of beta barrels that have not been exposed to lipids. A more balanced wording would be that other hemolysins were shown to interact with lipids at the rim domain. This was confirmed for VcaHL by the presence of a His cluster in the rim domain which, when mutated, lead to loss of binding. The legend of Fig. 6 should also mention if calcium was added and in what concentration. Regarding the statement on page 14, line 245, about fewer pre-pore complexes resulting from substitutions, it would be interesting to have a native PAGE here as well (see point 2).

Responses: We appreciate your feedback and agree that our initial descriptions concerning the noisy blobs around the transmembrane regions being lipids was inaccurate. Upon reflection, these noisy densities are likely contributed by the SMA nanodisc, given that they appear only around the transmembrane region. Past studies have similarly used such membrane-associated densities in cryo-EM maps to define the membrane boundary for VCC and other membrane proteins (*J Cell Biol.* (2021),

220, e202102035; *Bioinformatics* (2020), 36(8), 2595). Hence, we have revised our manuscript's descriptions of these SMA densities to prevent potential misinterpretation and to emphasize the criticality of the rim region that harbors the clustered histidine residues.

With regards to the observation of fewer pre-pore complexes for the H415A and H422A mutants in the negative-stain TEM experiments, we acknowledge that our original description was unclear, as these oligomers could represent mature pore complexes as well. Since these experiments were conducted on cholesterol-containing membranes, the ring-shaped oligomers likely integrated into the membranes, forming transmembrane pores. We have adjusted our description to more accurately represent these ring-shaped oligomers.

Thank you for pointing out the need for an adjustment in the figure legend of Fig. 6. We have now revised the legend to reflect the inclusion of calcium and its concentration, as per your suggestion. We acknowledge the importance of this detail, as calcium is a crucial factor in the formation of pre-pore and pore complexes.

Regarding the suggested addition of a native PAGE to verify the increased proportion of incomplete intermediates in these mutants, our existing native PAGE results (shown in Fig. 2C) did not detect any pre-pore complexes and intermediates. This suggests that native PAGE might not be the ideal method to detect the incomplete intermediates in these mutants. We look forward to exploring the cryo-EM structures of these intermediate states in future studies, which would deepen our understanding of how these point mutations influence membrane association and ring assembly.

11. Following the previous point, the discussion at pages 19 and 20 about lipid binding should be made less bold. While the quality of the map is overall good, claiming that the TM resolution is so high and reveals discernible lipids is an overstatement.

Responses: We acknowledge that our original statements regarding the histidine residues contacting lipid molecules were perhaps overstated. We have revised these sentences to emphasize the clustering of these residues in the rim region of V α HL, an area known to be involved in membrane association in several pore-forming toxins. We believe this more accurately communicates the importance of these residues in V α HL's membrane association. Furthermore, we have scaled back our discussion about the high resolution of the transmembrane region. The elevated resolution could result from a variety of factors and is not necessarily due to the use of SMALP. We have accordingly toned down the discussion on the improvement of transmembrane resolution in our revised manuscript.

12. Materials and methods. More details would be helpful regarding:
-plasmid construction - a table with primers, possibly a plasmid map.

- protein expression - the range of concentrations used in the stepwise imidazole gradient elution should be indicated.

Responses: We appreciate your comments and have now included a list of primers used in the study in Supplementary Table S3. With regard to the plasmid map for V α HL expression, it is based on pET24a. This map is readily available on the manufacturer's website, and given that the construct method and primers have been detailed in the materials and methods section, we believe a plasmid map may not add substantial information for the readers of this study. In relation to the protein expression and purification, we have updated the materials and methods section to include the range of concentrations used in the stepwise imidazole gradient elution.

13. Have the authors tried to improve the resolution of the beta barrel by masking out or subtracting the fuzzy densities of SMA density and of the beta-prism lectin domain?

Responses: We have not specifically masked out the SMA density in our study. As illustrated in our data analysis flowchart, the protein density could be effectively reconstituted without being disturbed by the fuzzy nanodisc region. This suggests the stability of V α HL within the SMALPs context.

Reviewers' Comments:

Reviewer #1:

Remarks to the Author:

This revised version of the manuscript is much improved according to both reviewers. I acknowledge that the authors have addressed all of the comments.

Considering the changes to the manuscript (in particular Fig 2) it is now more clear that what the authors are observing, indeed likely constitutes pre-pore or non-membrane inserted intermediate structures. This was not obvious in the original submission. I am now happy for the authors to suggest these oligomers are pre-pores.

Regarding my original comment:

Line 96 - A measurements between the EC 50 is not performed in any statistical manner. So it is not clear what is meant by significantly lower.

Response: We appreciate your comment on the need for statistical clarity in the presentation of our EC50 data. In response, we have now provided the 95% confidence intervals (CI) for the EC50 values in the manuscript. When comparing the EC50 for calcium-stimulated hemolytic activity with the control group lacking calcium ions, the 95% confidence intervals of the two groups do not overlap. This lack of overlap implies statistically significant differences. We thought this approach provides a more robust and statistically appropriate representation of our findings.

Acknowledged. To be clear, I was not requesting the authors perform statistical analysis re the assays, but rather pointing out that the original submission did not contain a stat. For clarity (and at the authors' discretion) in the text, it might be simpler if stated the relative difference e.g., 'The EC50 value of VcaHL decreases to 1.29 μ M compared to 33.84 μ M (a 26-fold decrease)'.

Reviewer #2:

Remarks to the Author:

The authors have addressed satisfactorily all the points that I raised. The quality of the manuscript has improved and it is good for publication in my view.

Point by point response to reviewer's comments:

Reviewer #1 (Remarks to the Author):

This revised version of the manuscript is much improved according to both reviewers. I acknowledge that the authors have addressed all of the comments.

Considering the changes to the manuscript (in particular Fig 2) it is now more clear that what the authors are observing, indeed likely constitutes pre-pore or non-membrane inserted intermediate structures. This was not obvious in the original submission. I am now happy for the authors to suggest these oligomers are pre-pores.

Response: We sincerely appreciate your thoughtful feedback on our revised manuscript and are grateful for your recognition of the improvements made. Your constructive feedback has played an instrumental role in enhancing the quality of our manuscript.

Regarding my original comment:

Line 96 - A measurements between the EC 50 is not performed in any statistical manner. So it is not clear what is meant by significantly lower.

Response: We appreciate your comment on the need for statistical clarity in the presentation of our EC50 data. In response, we have now provided the 95% confidence intervals (CI) for the EC50 values in the manuscript. When comparing the EC50 for calcium-stimulated hemolytic activity with the control group lacking calcium ions, the 95% confidence intervals of the two groups do not overlap. This lack of overlap implies statistically significant differences. We thought this approach provides a more robust and statistically appropriate representation of our findings.

Acknowledged. To be clear, I was not requesting the authors perform statistical analysis re the assays, but rather pointing out that the original submission did not contain a stat. For clarity (and at the authors' discretion) in the text, it might be simpler if stated the relative difference e.g., 'The EC50 value of VcaHL decreases to 1.29 uM compared to 33.84 uM (a 26-fold decrease)'.

Response: Thank you for your suggestion. We have further refined the phrasing to directly state the relative difference as you've recommended. The modified sentence now reads: "the EC50 value of VcaHL decreases to 1.29 μ M compared to 33.84 μ M (a 26-fold decrease) upon the addition of 10 mM CaCl₂" (Page 6, Line 98). We believe this change better emphasizes the magnitude of difference in a comprehensible manner.

Reviewer #2 (Remarks to the Author):

The authors have addressed satisfactorily all the points that I raised. The quality of the manuscript has

improved and it is good for publication in my view.

Response: Thank you for your positive feedback and constructive suggestions which have significantly improved the quality of our manuscript. We are pleased to know that the revised version meets your approval for publication.